# DESIGN-BENCH: BENCHMARKS FOR DATA-DRIVEN OFFLINE MODEL-BASED OPTIMIZATION

## ABSTRACT

Black-box model-based optimization (MBO) problems, where the goal is to find a design input that maximizes an unknown objective function, are ubiquitous in a wide range of domains, such as the design of drugs, aircraft, and robot morphology. Typically, such problems are solved by actively querying the black-box objective on design proposals and using the resulting feedback to improve the proposed designs. However, when the true objective function is expensive or dangerous to evaluate in the real world, we might instead prefer a method that can optimize this function using only previously collected data, for example from a set of previously conducted experiments. This data-driven offline MBO setting presents a number of unique challenges, but a number of recent works have demonstrated that viable offline MBO methods can be developed even for high-dimensional problems, using high-capacity deep neural network function approximators. Unfortunately, the lack of standardized evaluation tasks in this emerging new field has made tracking progress and comparing recent methods difficult. To address this problem, we present Design-Bench, a benchmark suite of offline MBO tasks with a unified evaluation protocol and reference implementations of recent methods. Our benchmark suite includes diverse and realistic tasks derived from real-world problems in biology, material science, and robotics that present distinct challenges for offline MBO methods. Our benchmarks, together with the reference implementations, are available at sites.google.com/view/design-bench. We hope that our benchmark can serve as a meaningful metric for the progress of offline MBO methods and guide future algorithmic development.

## 1 INTRODUCTION

Automatically synthesizing designs that maximize a desired objective function is one of the most important problems in many scientific and engineering domains. From protein design in molecular biology Shen et al. (2014) to superconducting material discovery in physics Hamidieh (2018), researchers have made significant progress in applying machine learning methods to such optimization problems over structured design spaces. Commonly, the exact form of the objective function is unknown, and the objective values for a novel design can only be evaluated by running either computer simulations or physical experiments in the real world. The process of optimizing an unknown function is known as black-box optimization, and is typically solved in an **online** iterative manner, where in each iteration the solver proposes new designs and query the objective function for feedback in order to propose better design in the next iteration Williams & Rasmussen (2006). In many domains however, the evaluation of the objective function is prohibitively expensive, because it requires manually conducting experiments in the real world. In this setting, one cannot simply query the true objective function to gradually improve the design. Instead, a collection of past records of designs and their corresponding objective values might be available, and therefore the optimization method must leverage the available data to synthesize the most optimal design possible. This is the setting of data-driven **offline model-based optimization**.

Although online black-box optimization has been studied extensively, the offline MBO problem has received comparatively less attention, and only a small number of recent works study offline MBO in the setting with high-dimensional design spaces, where they utilize deep learning techniques Brookes et al. (2019); Kumar & Levine (2019); Fannjiang & Listgarten (2020). This is partly due to the fact that methods for online design optimization cannot be easily applied in the offline

MBO setting. However, even with only a few existing methods, it is still hard to compare and track the progress in this field, as these methods are proposed and evaluated on different tasks with distinct evaluation protocols. To the best of our knowledge, there is no commonly adopted set of benchmarks for offline MBO. To address this problem, in this paper we introduce a suite of offline MBO benchmarks with standardized evaluation protocols. We include a realistic and diverse set of tasks that spans a wide range of application domains, from synthetic biology to robotics. The realism and diversity of the tasks is essential for the evaluation of offline data-driven model-based optimization methods, as it measures the generality of the methods being evaluated across multiple domains and verifies that they are not overfitting to a single task. Our benchmark tasks incorporate a variety of challenging factors, such as high dimensional input spaces and sensitive discontinuous objective functions, which help better identify the strengths and weaknesses of MBO methods.

Along with the benchmark suite, we also present reference implementations of a number of existing offline MBO methods and baselines. We systematically evaluate them on all of the proposed benchmark tasks and report our findings. A surprising discovery from our findings is that with proper data normalization, the simple baseline method of learning an objective value predictor and performing gradient ascent on its input outperforms several prior MBO methods in our benchmark. We hope that our work can provide insight into the current progress of offline MBO methods and can also serve as a meaningful metric to galvanize research in this area.

## 2 OFFLINE MODEL-BASED OPTIMIZATION PROBLEM STATEMENT

The goal in offline model-based optimization is to optimize an unknown (possibly stochastic) objective function $f(\mathbf{x})$, provided access to a static dataset $\mathcal{D} = \{(\mathbf{x}_i, y_i)\}$ of designs $\mathbf{x}_i$ and a corresponding measurement of the objective value $y_i$. Similar to batch Bayesian optimization (BBO) González et al. (2016), each algorithm $\mathfrak{A}$ is allowed to consume the dataset $\mathcal{D}$, and is required to produce a set of $K$ candidate designs $\mathfrak{A}(\mathcal{D}, K) = \{\mathbf{x}_i^* : i \in \{1...K\}\}$. These $K$ candidates are evaluated under the ground truth objective function $f(\mathbf{x})$, and the best performing design is reported as the final performance value. Abstractly, the objective for offline MBO is:

$$\arg \max_{\mathfrak{A}} \left[ \mathbb{P}(\{f(\mathbf{x}^*) : \mathbf{x}^* \in \mathfrak{A}(\mathcal{D}, K)\}, N) \right], \qquad (1)$$

where $\mathbb{P}$ denotes the percentile function. Intuitively, this formulation ranks an offline MBO algorithm using the $N^{\text{th}}$ percentile objective value obtained by it given a fixed evaluation budget of $K-$samples. Common choices of $N$ are 100, which represent the max objective value, and 50 which represents the median objective value among the candidates.

**What makes offline MBO especially challenging?** The offline nature of the problem requires that the algorithm $\mathfrak{A}$ not be tuned by peeking into the ground truth objective $f$, and this makes the offline MBO problem much more difficult than the online design optimization problem. One simple idea to tackle this problem is to learn an objective proxy using the dataset, and then convert this offline MBO problem into an online problem by treating the learned objective proxy as the true objective. However, this idea may not work well, due to the intrinsic out-of-distribution nature of optimal designs. First of all, in a number of practical MBO problems such as optimization over proteins or robot morphologies, the designs with highest objective values in the dataset already lie on the tail of the dataset distribution, since they are better than most other designs. In order to improve upon the best designs, an optimization method needs to produce designs that are even further away from the dataset distribution. For such out-of-distribution designs, is would be impossible to guarantee that the learned objective proxy is accurate, and hence any powerful optimization method would easily "exploit" the learned objective proxy and produce falsely promising designs that are drastically overestimated by the learned objective proxy. This conflict between the out-of-distribution nature of optimization and the in-distribution requirement of any learned model is indeed the core challenge of offline MBO. This challenge is often exacerbated in real-world problems by the high dimensionality of the design space and the sparsity of the available data, as we will show in our benchmark. A good offline MBO method needs to carefully balance the two sides of the conflict, producing optimized designs that are not too far from the data distribution.

| Task | Dataset Size | Design Dimensions | Design Space | Discrete Categories |
|------|-------------|-------------------|--------------|---------------------|
| **GFP-v0** | 5000 | 238 | Discrete | 20 |
| **GFP-v1** | 6852 | 238 | Discrete | 30 |
| **MoleculeActivity-v0** | 4216 | 1024 | Discrete | 2 |
| **Superconductor-v0** | 16953 | 81 | Continuous | N/A |
| **HopperController-v0** | 3200 | 5126 | Continuous | N/A |
| **AntMorphology-v0** | 12300 | 60 | Continuous | N/A |
| **DKittyMorphology-v0** | 9546 | 56 | Continuous | N/A |

Table 1: **Overview of the tasks in our benchmark suite.** Design-Bench includes a variety of tasks from different domains with both discrete and continuous design spaces and 3 high-dimensional tasks with $> 200$ design dimensions, making it suitable for benchmarking offline MBO methods.

## 3 RELATED WORK

While the extensive prior work on MBO, particularly Bayesian optimization, has made tremendous progress both in terms of enabling intelligent active selection of query points Kolda et al. (2003); Lizotte (2008); Rubinstein & Kroese (2013); Powell (1998); Whitley (1994) and scaling up the dimensionality of Bayesian optimization methods Snoek et al. (2012; 2015); Shahriari et al. (2015), a significant number of real-world problems may be approached more naturally as offline MBO problems, given the availability of several datasets which contain designs annotated with their corresponding score values. Offline MBO presents a new set up challenges, as we discuss below, and we believe that these challenges require a new set of benchmarks that emphasize offline data and high-dimensional design spaces.

Researchers working on either online active design optimization or offline model-based optimization have collected various datasets of designs that can be used to build tasks for offline MBO. Sarkisyan et al. (2016) analyze the fluorescence of GFP proteins under blue and ultraviolet light, and Brookes et al. (2019) use this dataset for optimization to find the protein with the highest fluorescence value. ChEMBL Gaulton et al. (2012) provides a dataset for drug discovery, where molecule activities are measured against a target assay. Hamidieh (2018) analyze the critical temperatures for superconductors and provide a dataset to search for room-temperature superconductors with potential in the construction of quantum computers. Some of these datasets have already been employed in the study of offline MBO methods Kumar & Levine (2019); Brookes et al. (2019); Fannjiang & Listgarten (2020). However, these studies use different set of tasks and their evaluation protocols are highly field-specific, making it difficult to use for general algorithm development. In our benchmark, we incorporate modified variants of some of these datasets along with our own tasks and provide a standardized evaluation protocol. We hope that these tasks can represent realistic MBO problems across a wide range of domains and that the standardized evaluation protocol can facilitate development of new and more powerful offline MBO algorithms.

Recently, researchers have proposed several methods specifically for the completely offline MBO problem. To make up for the fact that the true objective function is not available, these methods often make use of some type of learned objective function as a proxy for the real one and a generative model Kingma & Welling (2013); Goodfellow et al. (2014) to capture the distribution of valid designs, especially for high dimensional design spaces. Kumar & Levine (2019) learns a objective value conditioned generative model to synthesize the designs with the desired objective value. Brookes et al. (2019); Fannjiang & Listgarten (2020) employ a variational autoencoder Kingma & Welling (2013) to capture the distribution of designs in the dataset and a learned neural network objective function model. In our benchmark, we provide open-source reference implementations for these methods and systematically evaluate them in all our tasks.

## 4 DESIGN-BENCH BENCHMARK TASKS

In this section, we describe the set of tasks included in our benchmark. We first provide an overview of the tasks in Table 1. Each task in our benchmark suite comes with a dataset $\mathcal{D} = \{(\mathbf{x}_i, y_i)\}$, along with an oracle objective function $f$ which is to be utilized for for test time evaluation only. An offline MBO algorithm should not query the oracle during training time, even for hyperparameter tuning. While in some of the tasks in our benchmark, such as tasks pertaining to robotics –

HopperController-v0, DKittyMorphology-v0 and AntMorphology-v0, the oracle functions are evaluated by running computer simulations to obtain the true objective values, in the other tasks, the true objective values can only be obtained by conducting expensive physical experiments. While the eventual aim of offline MBO is to make it possible to optimize designs in precisely such settings where querying the groundtruth objective is challenging, requiring real physical experiments for evaluation makes the design and benchmarking of new algorithms difficult and time consuming. Therefore, to facilitate evaluation, following evaluation methodology in prior work Brookes et al. (2019); Fannjiang & Listgarten (2020), and use models built by experts as our oracle function. Such models are typically *also* learned models, but with representations that are hand-designed, built-in domain-specific inductive biases and typically trained on much more data than is made available for solving the offline MBO problem, which increases the chance that this proxy "true function" can answer queries outside of the MBO training distribution. While this approach to evaluation diminishes the realism of our benchmark, since these proxy "true functions" may not always be accurate, we believe that this trade off is worth it to make benchmarking practical. The main purpose of our benchmark is to facilitate the evaluation and development of offline MBO algorithms, and we believe that it is important to include tasks in domains where the true objective values can only be obtained via physical experiments, which make up a large portion of the real-world MBO problems. For each task we discuss in this section, we provide a detailed description of the data collection strategy and the data pre-processing strategy in Appendix A.

**GFP-v0 and GFP-v1: protein fluorescence maximization.** The goal of this task is to design a derivative protein from the *Aequorea victoria* green fluorescent protein (GFP) that has maximum fluorescence, using a real-world dataset mapping proteins to fluorescence collected by Sarkisyan et al. (2016). While we cannot exactly evaluate any novel protein, we employ an expert Gaussian process regression model with a protein-specific kernel built by Shen et al. (2014) as the oracle function, following the convention of Brookes et al. (2019); Fannjiang & Listgarten (2020). This expert model is built on a larger dataset than the one we use in this benchmark, making it relatively accurate for proteins not in our dataset. The design space for GFP is discrete, consisting of a sequence of 238 categorical variables that can take one of 20 or 30 options, which corresponds to the choice of 238 amino acids that make up the green fluorescent protein from 20 types of amino acid. In brief, GFP-v0 uses a Gaussian process as the expert model, while GFP-v1 adapts the 12-layer Transformer provided by the TAPE framework as the expert model. More details are in Appendix A.

**MoleculeActivity-v0: drug activity maximization.** This task is taken from the domain of drug discovery, where the goal is to design the substructure of a molecule that exhibits high activity when tested against a target assay. The dataset we provide with this task was originally collected by ChEMBL Gaulton et al. (2012). Again like for the GFP task, the true molecule activity against a target assay can only be evaluated with physical experiments. Therefore, we leverage an oracle used recently for meta-learning Yao et al. (2020) and adopt the expert random forest regression model built by Martin et al. (2019) as the oracle function. From the ChEMBL data, we pick a single target assay to form the dataset of this task, resulting in 4216 molecules in total. The design space for this task is a sequence of 1024 binary categorical variables that correspond to the Morgan radius 2 substructure fingerprints, making this task a high-dimensional task.

**Superconductor-v0: critical temperature maximization for superconductor materials.** The Superconductor-v0 task is taken from the domain of materials science, where the goal is to design a superconducting material that has a high critical temperature. Developing superconducting materials has immense practical value, especially for room-temperature superconductors, which could greatly facilitate quantum computing. We adapt a real-world dataset used by materials scientists Hamidieh (2018). The dataset contains 21263 superconductors annotated with critical temperatures. Prior work has employed this dataset for the study of offline MBO methods Brookes et al. (2019), and we follow their convention of using an expert random forest model, detailed in Fannjiang & Listgarten (2020), for our oracle function. The design space for Superconductor-v0 is an 81-channel vector of continuous variables, which describe properties of the material such as its atomic radius.

**HopperController-v0: robot neural network controller optimization.** HopperController-v0 is a task created by us in the domain of robotics, where the goal is to optimize weights of a neural network policy that maximizes expected return on the Hopper-v2 task in OpenAI Gym Brockman et al. (2016). While this goal might look similar to that of reinforcement learning, it is important to note that the formulation is entirely different. Unlike reinforcement learning, we don't have

access to any form of trajectory data, neither by actively sampling from the environment nor from a pre-collected dataset. Instead, we only have a dataset of weights of a neural network controller 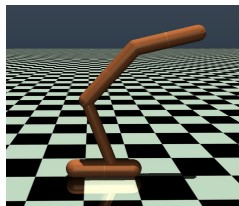 and the corresponding return values. Therefore, reinforcement learning methods cannot be applied to solve this problem. Unlike tasks introduced before, we evaluate the true objective value of any design during dataset generation by running 1000 steps of simulation in the MuJoCo simulator. This horizon length is chosen so that the learning problem is long-horizon, and difficult. The design space of this task is high-dimensional with 5126 continuous variables corresponding to the flattened weights of a neural network controller. The dataset is collected by training a PPO reinforcement learning agent Schulman et al. (2017) and recording the agent's weights every 10,000 samples.

**AntMorphology-v0 and DKittyMorphology-v0: robot morphology optimization.** We created these two tasks to optimize the morphological structure of two simulated robots: Ant from OpenAI Gym Brockman et al. (2016) and D'Kitty from ROBEL Ahn et al. (2019). For AntMorphology, the goal is to optimize the morphology of an ant shaped robot, to run as fast as possible, with a pre-trained neural network controller. For DKittyMorphology, the goal is to optimize the morphology of D'Kitty, a quadrupedal 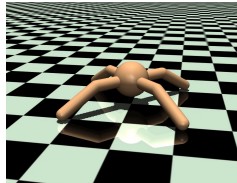 robot, such that a pre-trained neural network controller can navigate the robot to a fixed location. In this fashion, the goal for both tasks is to *recover* a morphology with which the pre-trained controller is compatible. The variable morphology parameters of both robots include size, orientation, and location of the limbs, giving us 60 continuous values in total for Ant and 56 for D'Kitty. To evaluate the ground truth value for a given design, we run robotic simulation in MuJoCo Todorov et al. (2012) for 100 time steps, averaging 16 independent trials. These parameters are chosen to reduce stochasticity, and allow the simulator to run in a minimal amount of time.

## 5 TASK PROPERTIES, CHALLENGES, AND CONSIDERATIONS

The primary goal of our proposed benchmark is to provide a general test bench for developing, evaluating, and comparing algorithms for offline MBO. While in principle any online active design optimization problem can be formulated into an offline MBO problem by collecting a dataset of designs and corresponding objective measurements, it is important to pick a subset of tasks that represent the challenges of real-world MBO problems in order to convincingly evaluate algorithms and obtain insights about algorithm behavior. Therefore, several factors must be considered when choosing the tasks, which we discuss next.

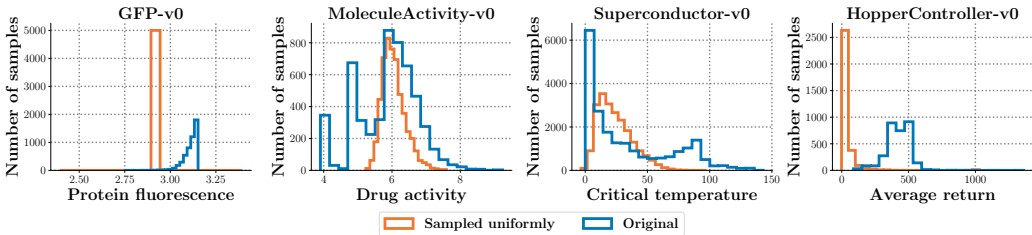

Figure 1: **Histogram (frequency distribution) of objective values in the dataset compared to a uniform re-sampling of the dataset** from the design space. In every case, re-sampling skews the distribution of scores to the left, suggesting that there exists a thin manifold of valid designs in the high-dimensional design space, and most of the volume in this space is occupied by low-scoring designs. The distribution of objective values in the dataset are often heavy-tailed, for instance, in the case of MoleculeActivity-v0 and Superconductor-v0.

**Diversity and Realism.** First of all, the tasks need to be diverse and realistic in order to prevent offline MBO algorithms from overfitting to a particular problem domain and to expect that offline MBO methods performing well on this benchmark suite would also perform well on other real-world offline MBO problems. Design-Bench consists of tasks diverse in many respects: it includes both tasks with *discrete* and with *continuous* design spaces, which have distinct implications for offline MBO algorithms. Continuous design spaces, equipped with metric space and ordering structures, could make the problem easier to solve than discrete design spaces. On the other hand, discrete design spaces are finite and therefore the dataset coverage might be better than some continuous

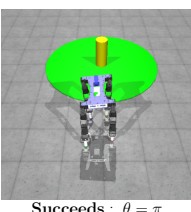 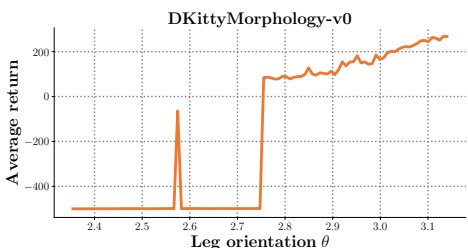 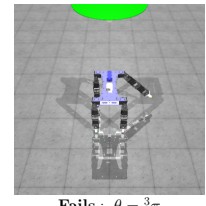

Succeeds : $\theta = \pi$      Fails : $\theta = \frac{3}{4}\pi$

Figure 2: **Highly sensitive landscape of the groundtruth objective function in DKittyMorphology-v0.** A small change in a single dimension of the design-space, for instance, by changing the orientation $\theta$ (x-axis) of the base of the agent's front right leg, the performance value (y-axis) of the agent is critically impacted. The agent's design is the original D'Kitty design and is held constant while varying $\theta$ uniformly from $\frac{3}{4}\pi$ to $\pi$.

tasks. A strong offline MBO algorithm needs to be able to handle both cases. Moreover, our tasks have varying dimensionality, ranging from 56 to 5126 dimensions. And finally, our tasks span multiple realistic domains ranging from biology and physical science to robotics.

**High-dimensional design space.** In many real-world offline MBO problems such as drug discovery Gaulton et al. (2012), the design space is *high-dimensional* and the valid designs lie on a *thin manifold* in this high-dimensional space. Due to the curse of dimensionality, the available design data is often sparsely distributed in the design space. This property poses a unique challenge for many MBO algorithms: to be effective on such problem domains, MBO methods need to strongly restrict optimization to a thin manifold of the design space to be able to produce valid designs. Prior work Kumar & Levine (2019) has noted that this can be extremely hard in several cases. In our benchmark, we include the GFP-v0, MoleculeActivity-v0 and HopperController-v0 tasks with high-dimensional design spaces to capture this challenge. To intuitively understand this challenge, we performed a study on the HopperController-v0 task in Figure 1, where we sampled 3200 designs uniformly at random from the design space and plotted a histogram of their objective values against those in the dataset we provide, that only consists of valid designs. Observe the clear discrepancy in the objective values, where randomly sampled designs attain objective values much lower than the dataset. This indicates that valid designs only lie on a thin manifold in the design space and therefore we are very unlikely to hit a valid design by uniform random sampling.

**Highly sensitive objective function.** Another important challenge that should be taken into consideration is the *high sensitivity* of objective functions, where closeness of two designs in design space need not correspond to closeness in their objective values, which may differ drastically. This challenge is naturally present in practical problems like protein synthesis Shen et al. (2014), where the change of a single amino acid could significantly alter the property of the protein. The DKittyMorphology-v0 and AntMorphology-v0 tasks in our benchmark suite are also particularly challenging in this direction. To visualize the high sensitivity of the objective function, we plot a one dimensional slice of the objective function around a single sample in our dataset in Figure 2. We see that with other variables kept the same, slightly altering one variable can significantly reduce the objective value, making it hard for offline MBO methods to produce the optimal design.

**Heavy-tailed data distributions.** Another important property that can pose a challenge for offline MBO methods is the shape of the data distribution. In particular, machine learning algorithms are likely to exhibit poor learning behavior when the distribution of objective values in the dataset is heavy-tailed. This challenge is often present in black-box optimization (Chowdhury & Gopalan, 2019) and can hurt the performance of MBO algorithms that use a generative model as well as those that use a learned model of the objective function. As shown in Figure 1, the Superconductor-v0 and MoleculeActivity-v0 tasks in our benchmark exhibit this heavy-tailed structure.

## 6    ALGORITHM IMPLEMENTATIONS IN DESIGN-BENCH

To provide a baseline for comparisons in future work, we ran a number of recently proposed MBO algorithms on each of our tasks. Since the dimensionality of our tasks ranges from 56 to 5126, we chose methods that have been shown in prior work to handle *both* the case of offline training data (i.e., no active interaction) and high-dimensional inputs. To that end, we include MINs Kumar & Levine (2019), CbAS Brookes et al. (2019), and autofocusing CbAS Fannjiang & Listgarten (2020) in our comparisons, along with a baseline naïve "gradient ascent" method that approximates the

true function $f(\mathbf{x})$ with a deep neural network and then performs gradient ascent on the output of this model. In this section, we briefly discuss these algorithms, before performing a comparative evaluation in the next section.

**Gradient Ascent.** We evaluate a simple baseline that learns a model of the objective function, $\hat{f}(\mathbf{x})$ and optimizes $\mathbf{x}$ against this learned model via gradient ascent. Formally, the optimal solution $\mathbf{x}^*$ generated by this method can be computed as a fixed point of the following update: $\mathbf{x}_{t+1} \leftarrow \mathbf{x}_t + \alpha \nabla_{\mathbf{x}} \hat{f}(\mathbf{x})|_{\mathbf{x}=\mathbf{x}_t}$. In practice we perform $T$ gradient steps, and report $\mathbf{x}_T$ as the final solution. Such methods are notoriously susceptible to falling off-the-manifold of valid inputs since nothing constrains the resulting $\mathbf{x}_T$ to be on the manifold of valid-designs. We employ three variants of this procedure, the first optimizing a single learned objective function $\hat{f}(\mathbf{x})$, the second (**Grad. Min**) optimizing over the minimum prediction of $N$ learned objective functions in an ensemble, and the third (**Grad. Mean**) optimizing over the mean ensemble prediction. We set $N = 5$ in our experiments. We provide details of the gradient ascent procedure in the Appendix Section E.

**Conditioning by adaptive sampling (CbAS).** CbAS learns a density model in the space of design inputs, $p_0(\mathbf{x})$ that approximates the data distribution and gradually adapts it towards the optimized solution $\mathbf{x}^*$. In a particular iteration $t$, CbAS alternates between **(1)** training a variational auto-encoder (VAE) Kingma & Welling (2013) on a set of samples generated from the previous model $\mathcal{D}_t = \{\mathbf{x}_i\}_{i=1}^m; \mathbf{x}_i \sim p_{t-1}(\cdot)$ using a weighted version of the standard ELBO objective that puts a higher likelihood on *estimated* high-scoring designs (i.e., designs that have a score beyond a certain pre-specified threshold), and **(2)** generating new design samples from the autoencoder to serve as $\mathcal{D}_{t+1} = \{\mathbf{x}_i | \mathbf{x}_i \sim p_t(\cdot)\}$. In order to estimate the objective values for design samples from the learned density model $p_t(\mathbf{x})$, CbAS utilizes separately trained models of the objective function, $\hat{f}(\mathbf{x})$, that are trained via supervised regression to map an input $\mathbf{x}_i$ to its objective value $y_i$. This training process, at a given iteration $t$, can formally be described as:

$$p_{t+1}(\mathbf{x}) := \arg\min_p \frac{1}{m} \sum_{i=1}^m \frac{p_0(\mathbf{x}_i)}{p_t(\mathbf{x}_i)} P(\hat{f}(\mathbf{x}_i) \geq \tau) \log p_t(\mathbf{x}_i), \text{ where } \{\mathbf{x}_i\}_{i=1}^m \sim p_t(\cdot). \quad (2)$$

**Autofocusing CbAS.** Since CbAS uses a learned model of the objective function $\hat{f}(\mathbf{x})$ as a proxy for the ground-truth function to iteratively adapt the generative model $p(\mathbf{x})$ towards the optimized design, the function $\hat{f}(\mathbf{x})$ will inevitably be required to make predictions under shifting design distributions $p_t(\mathbf{x})$. Since these learned predictions are used as ground-truth values, any inaccuracy in the learned function can adversely affect the optimization procedure. Autofocused CbAS aims to correct for this distribution shift by re-training $\hat{f}(\mathbf{x})$ (now denoted $\hat{f}_t(\mathbf{x})$) under the design distribution given by the current model, $p_t(\mathbf{x})$ via importance sampling, which is then used by CbAS.

$$\hat{f}_{t+1} := \arg\min_{\hat{f}} \frac{1}{|\mathcal{D}|} \sum_{i=1}^{|\mathcal{D}|} \frac{p_t(\mathbf{x}_i)}{p_0(\mathbf{x}_i)} \cdot \left(\hat{f}(\mathbf{x}_i) - y_i\right)^2,$$

**Model inversion networks (MINs).** MINs learn an inverse map from the objective value to a design, $\hat{f}^{-1} : \mathcal{Y} \to \mathcal{X}$ by using objective-conditioned inverse maps, search for optimal $y$ values during optimization and finally query the learned inverse map to produce the corresponding optimal design. These methods minimize a divergence measure $\mathcal{L}_p(\mathcal{D}) := \mathbb{E}_{y \sim p_{\mathcal{D}}(y)} \left[ D(p_{\mathcal{D}}(\mathbf{x}|y), \hat{f}^{-1}(\mathbf{x}|y)) \right]$, to train such an inverse map. During optimization, MINs obtain the optimal $y$-value that is used to query the inverse map, and obtains the optimized design by sampling form the inverse map.

**Covariance matrix adaptation (CMA-ES).** CMA-ES Hansen (2006) is an evolutionary algorithm that maintains a distribution over the optimal design, and gradually refines this distribution by adapting the covariance matrix. Formally, let $\mathbf{x}_t \sim \mathcal{N}(\mu_t, \Sigma_t)$ be the samples obtained from the Gaussian distribution at an iteration $t$ of the algorithm, then CMA-ES computes the value of a learned objective function $\hat{f}(\mathbf{x}_t)$ on samples $\mathbf{x}_t$, and fits $\Sigma_{t+1}$ to the highest scoring fraction of samples $\mathbf{x}_t$ and iterates the process for multiple iterations. Typically CMA-ES utilizes the groundtruth objective function, but as is evident from above, we utilize a learned model of the objective in the offline setting. This learned model, $\hat{f}(\mathbf{x})$ is learned via supervised regression.

**REINFORCE (Williams, 1992).** (Adaptation of Angermueller et al. (2020)) We also evaluated a method that aims to optimize a learned objective function, $\hat{f}(\mathbf{x})$, using the REINFORCE-style

| Algorithm Name | GFP-v0 | GFP-v1 | MoleculeActivity-v0 | Superconductor-v0 |
|---|---|---|---|---|
| **Autofocus** | $1.531 \pm 0.057$ (0.031) | $1.580 \pm 0.001$ (0.000) | $0.920 \pm 0.053$ (0.029) | $1.041 \pm 0.150$ (0.083) |
| **CbAS** | $1.638 \pm 0.072$ (0.039) | $1.580 \pm 0.001$ (0.000) | $0.903 \pm 0.049$ (0.027) | $0.975 \pm 0.117$ (0.064) |
| **MINs** | $1.406 \pm 0.082$ (0.046) | $1.580 \pm 0.001$ (0.000) | $0.981 \pm 0.089$ (0.049) | $1.084 \pm 0.144$ (0.079) |
| **Gradient Ascent** | $0.356 \pm 0.002$ (0.000) | $-1.775 \pm 0.010$ (0.005) | $1.029 \pm 0.025$ (0.034) | $1.211 \pm 0.124$ (0.070) |
| **Grad. Min** | $0.441 \pm 0.185$ (0.102) | $-1.670 \pm 0.231$ (0.127) | $0.993 \pm 0.028$ (0.015) | $1.128 \pm 0.121$ (0.066) |
| **Grad. Mean** | $0.431 \pm 0.097$ (0.053) | $-1.286 \pm 1.017$ (0.560) | $0.987 \pm 0.044$ (0.024) | $1.189 \pm 0.141$ (0.078) |
| **REINFORCE** | $1.249 \pm 0.121$ (0.000) | $1.576 \pm 0.003$ (0.005) | $0.965 \pm 0.028$ (0.000) | $1.170 \pm 0.058$ (0.000) |
| **BO-qEI** | $0.999 \pm 0.000$ (0.000) | $1.579 \pm 0.000$ (0.000) | $1.222 \pm 0.062$ (0.000) | $1.213 \pm 0.000$ (0.000) |
| **CMA-ES** | $0.356 \pm 0.000$ (0.000) | $-1.755 \pm 0.006$ (0.018) | $1.081 \pm 0.036$ (0.000) | $1.332 \pm 0.071$ (0.000) |

Table 2: **100th percentile** evaluations on the first four tasks in our benchmark. Results are averaged over 16 trials, the $\pm$ indicates the standard deviation of the reported performance, and the parenthesis at the end of each cell indicate the **95th confidence interval** of the sample mean by fitting a T distribution to the performance sample points, calculated using the T distribution from `scipy.stats`.. Values are normalized to a zero if the solution found by optimization performs like the worst sample in the training dataset, and a one if the solution performs like the best sample in the training dataset. A value $> 1$ indicates that a solution better that the best observed sample is found. Similarly, a value $< 0$ indicates that the optimizer has found a solution worse that the worst observed sample. Unnormalized results can be found in Table 6 in Appendix D

| Algorithm Name | HopperController-v0 | AntMorphology-v0 | DKittyMorphology-v0 |
|---|---|---|---|
| **Autofocus** | $0.326 \pm 0.105$ (0.058) | $2.158 \pm 0.044$ (0.074) | $1.787 \pm 0.233$ (0.428) |
| **CbAS** | $0.402 \pm 0.311$ (0.171) | $2.183 \pm 0.016$ (0.029) | $1.752 \pm 0.298$ (0.547) |
| **MINs** | $0.548 \pm 0.468$ (0.257) | $2.164 \pm 0.038$ (0.021) | $1.672 \pm 0.190$ (0.104) |
| **Gradient Ascent** | $0.772 \pm 0.209$ (0.100) | $2.212 \pm 0.021$ (0.040) | $1.858 \pm 0.242$ (0.322) |
| **Grad. Min** | $1.434 \pm 0.327$ (0.180) | *running* | *running* |
| **Grad. Mean** | $1.379 \pm 0.517$ (0.285) | *running* | *running* |
| **REINFORCE** | $0.406 \pm 0.217$ (0.000) | *running* | *running* |
| **BO-qEI** | $0.558 \pm 0.105$ (0.000) | *running* | *running* |
| **CMA-ES** | $0.497 \pm 0.253$ (0.000) | *running* | *running* |

Table 3: **100th percentile** evaluations on the final three tasks in our benchmark. Results are averaged over 16 trials, and the $\pm$ indicates the standard deviation of the reported performance. For a description of the score normalization methodology, refer to the caption of Table 2. Unnormalized results corresponding to this table can be found in Table 7 in Appendix D. "*running*" indicates that these runs have not yet completed

policy-gradient estimator. This estimator parameterizes a distribution $\pi_\theta(\mathbf{x})$ over the input space and then updates the parameters $\theta$ of this distribution towards the design that maximizes $\hat{f}(\mathbf{x})$, using the gradient, $\mathbb{E}_{\mathbf{x} \sim \pi_\theta(\mathbf{x})}[\nabla_\theta \log \pi_\theta(\mathbf{x}) \cdot \hat{f}(\mathbf{x})]$. This method can be regarded as an adaptation of the DynaPPO method from Angermueller et al. (2020) to our setting. Not all our tasks can be formulated as a sequential decision-making problem, therefore we pose offline MBO as a "one-step" problem. To mimic the selection of models from Angermueller et al. (2020), we train an ensemble of $\hat{f}(\mathbf{x})$ models and pick the subset of models that satisfy a validation loss threshold $\tau$. This threshold is task-specific; for example, $\tau \leq 0.25$ is sufficient for Superconductor-v0.

**Bayesian optimization (BO-qEI).** We perform offline Bayesian Optimization to maximize the value of a learned objective function, $\hat{f}(\mathbf{x})$, by fitting a Gaussian Process, proposing candidate solutions, and labelling these candidates using $\hat{f}(\mathbf{x})$. To improve efficiency, we choose the quasi-Expected-Improvement acquisition function Wilson et al. (2017), and the implementation of Bayesian Optimization provided in the BoTorch framework Balandat et al. (2019).

## 7 BENCHMARKING PRIOR METHODS

In this section, we provide a comparison of prior algorithms discussed in Section 6 on our proposed tasks. For purposes of standardization, easy benchmarking, and future algorithm development, we present results for all Design-Bench tasks in Table 2 and 4. As discussed in Section 2, we provide each algorithm with a dataset, and allow the method to produce $K = 128$ optimized design candidates. These $K = 128$ candidates are then evaluated with the oracle function, and we report the 100th percentile and the 50th percentile performance among them averaged over 16 independent trials, following the convention of prior offline MBO works Fannjiang & Listgarten (2020).

**Algorithm setup and hyperparameter tuning.** Since our goal is to generate high-performing solutions without *any* knowledge of the groundtruth, oracle function, any form of hyperparameter tuning on the parameters of the learned model should respect the evaluation boundary. We provide a recommended method for tuning each algorithm described in Section 6 that is fully offline. To briefly summarize, for **CbAS**, tuning amounts to finding a stable configuration for a VAE, such that samples

from the prior distribution map to on-manifold designs after reconstruction. We empirically found that a $\beta$-VAE was essential for stability of CbAS—and high values of $\beta > 1$ are especially important for modelling high-dimensional spaces like that of HopperController-v0. As a general task-agnostic principle for selecting $\beta$, we choose the smallest $\beta$ such that the VAE's latent space does not collapse during importance sampling. Collapsing latent-spaces seem to coincide with diverging importance sampling, and the VAE's reconstructions collapsing to a single mode. For **MINs**, tuning amounts to fitting a good generative model. We observe that MINs is particularly sensitive to the scale of $y_i$ when conditioning, which we resolve by normalizing the objective values. We implement MINs using WGAN-GP, and find that similar hyperparameters work well-across domains. For **Gradient Ascent**, while prior work has generally obtained extremely poor performance for naïve gradient-ascent based optimization procedures on top of learned models of the objective function, we find that by normalizing the designs $\mathbf{x}$ and objective values $y$ to have unit Gaussian statistics, and by multiplying the learning rate $\alpha \leftarrow \alpha\sqrt{d}$ where $d$ is the dimension of the design space (discussed in Appendix E), a naïve gradient ascent based procedure can perform reasonably well on most tasks, without any task-specific tuning. For discrete tasks, only the objective values are normalized, and optimization is performed over log-probabilities of designs. We then uniformly evaluate samples obtained by running 200 steps of gradient ascent starting from the highest scoring 128 samples in each dataset. Tuning instructions for each baseline in this work is available in Appendix F.

**Results and discussion.** The results for all tasks are provided in Table 2 and 4. There are several interesting takeaways from these results. First, these results indicate that there is no clear winner between the three prior methods MINs, CbAS and Auto-focused CbAS, provided they are all trained offline with no access to groundtruth evaluation for any form of hyperparameter tuning. Second, somewhat surprisingly, the naïve gradient ascent baseline is competitive with several highly sophisticated MBO methods in **5 out of 7** tasks (Table 2), especially on high-dimensional tasks (e.g., HopperController-v0). This result suggests that it might be difficult for generative models to capture high-dimensional task distributions

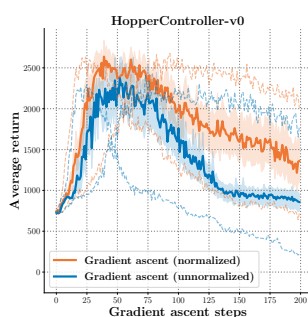

with enough precision to be used for optimization. We conducted an ablation study to determine the reasons behind the surprisingly good performance of the naïve baseline. We found that the identical gradient-ascent baseline performed a factor **1.4x** worse on HopperController-v0, when optimizing in the space of unnormalized designs and objective values, indicating that normalization is key in obtaining good performance with a naïve gradient ascent baseline. For additional information on how normalization is performed, refer to Appendix Section E.

Finally, we remark that the performance numbers for certain methods reported in Table 2 differ from the performance reported by prior works on certain tasks This difference stems from the standardization procedure employed in dataset generation (which we discuss in Appendix A), and the usage of uniform hyperparameters to ensure task-agnostic hyperparameter selection.

## 8 CONCLUSION

While online model-based optimization methods have received a lot of attention, especially in the Bayesian optimization community, offline model-based optimization is increasingly gaining popularity as it carries the promise to be able to convert existing databases of designs into powerful optimizers, without the need for any expensive online interaction. However, due to the lack of standardized benchmarks and evaluation protocols, it has been difficult to accurately track the progress in this field. To address this problem, we introduce Design-Bench, a benchmark suite of offline MBO tasks that covers a wide variety of domains, and both continuous and discrete, low and high dimensional design spaces. We provide a comprehensive evaluation of existing methods under identical assumptions. The somewhat surprising efficacy of simple baselines such as naïve gradient ascent suggests the need for careful tuning and standardization of methods in this area. Another interesting avenue for future work is to devise methods that can be used to perform model-selection and hyperparameter selection. One approach to address this problem is to devise methods for offline evaluation of produced solutions, which is also an interesting topic for future work. We hope that our benchmark will be adopted as the standard metric in evaluating offline MBO algorithms and provide meaningful insight in future algorithmic development.

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

# Appendices

## A  DATA COLLECTION

In this section, we detail the data collection steps used for creating each of the tasks in design-bench. We answer (1) where is the data from, and (2) what pre-processing steps are used?

### A.1  GFP-v0

The GFP task provided is a derivative of the GFP dataset Sarkisyan et al. (2016). The dataset we use in practice is that provided by Brookes et al. (2019) at the url `https://github.com/dhbrookes/CbAS/tree/master/data`. We process the dataset such that a single training example consists of a protein represented as a tensor $x_{\text{GFP}} \in \{0, 1\}^{238 \times 20}$. This tensor is a sequence of 238 one-hot vectors corresponding to which amino acid is present in that location in the protein. We use the dataset format of Brookes et al. (2019) with no additional processing. The data was originally collected by performing laboratory experiments constructing proteins similar to the Aequorea victoria green fluorescent protein and measuring fluorescence.

### A.2  GFP-v1

This variant of the GFP task provided is a derivative of the GFP dataset Sarkisyan et al. (2016). The dataset we use in practice is that provided by Brookes et al. (2019) at the url `https://github.com/dhbrookes/CbAS/tree/master/data`. We process the dataset such that a single training example consists of a protein represented as a tensor $x_{\text{GFP}} \in \{0, 1\}^{238 \times 30}$. This tensor is a sequence of 238 one-hot vectors corresponding to which amino acid is present in that location in the protein. We use the dataset format of Brookes et al. (2019) with no additional processing. The data was originally collected by performing laboratory experiments constructing proteins similar to the Aequorea victoria green fluorescent protein and measuring fluorescence.

### A.3  MOLECULEACTIVITY-V0

The MoleculeActivity task is a derivative of a much larger dataset that is derived from ChEMBL Gaulton et al. (2012), a large database of chemicals and their properties. The data, similar to GFP, was originally collected by performing physical experiments on a large number of molecules, and measuring their activity with a target assay. We have processed the original dataset presented in Martin et al. (2019), which consists of more than one million molecules and 11,000 assays, into a smaller scale task with 4216 molecules and a single assay. We select this assay by first calculating the number of unique scores present in the dataset per assay, and sorting the assays by the number of unique scores. We select assay 600885 from Martin et al. (2019). This particular assay has 4216 molecules after pre-processing. Our pre-processing steps include converting each molecule into a one-hot tensor $x_{\text{Molecule}} \in \{0, 1\}^{1024 \times 2}$. This is performed by calculating the Morgan radius 2 substructure fingerprints of 1024 bits, which is implemented in RDKit. This calculation requires the SMILES representation for each molecule, which is provided by Martin et al. (2019). The final step of pre-processing, is to sub sample the dataset by defining a percentile used to select and discard high-performing molecules, such that difficulty of the task is artificially increased. We use a split percentile of 80 for MoleculeActivity in the experiments in this paper.

### A.4  SUPERCONDUCTOR-V0

Superconductor-v0 is inspired by recent work Fannjiang & Listgarten (2020) that applies offline MBO to optimize the properties of superconducting materials for high critical temperature. The data we provide in our benchmark is real-world superconductivity data originally collected by Hamidieh (2018), and subsequently made available to the public at the url `https://archive.ics.uci.edu/ml/datasets/Superconductivty+Data#`. The original dataset consists of superconductors featurized into vectors $x_{\text{Superconductor}} \in \mathcal{R}^{81}$. One issue with the original dataset is that the largest value of a single dimension in the dataset is 22590.0, which appears to cause learning

instability. We follow Fannjiang & Listgarten (2020) and normalize each dimension of the design-space to have zero mean and unit variance. However, we deviate from the remaining pre-processing steps in Fannjiang & Listgarten (2020). In order to promote task realism, we directly use the super-conductivity data, whereas Fannjiang & Listgarten (2020) re-samples by collecting iid unit gaussian samples and labelling them with the task oracle function. This causes the scores in the dataset to correspond exactly to the scores provided by the oracle. No other domain in design-bench re-samples nor re-labels static data, so we omit it here for consistency.

### A.5   HOPPERCONTROLLER-V0

The HopperController task is one that we provide ourselves. The goal of this task is to design a set of weights for as neural network policy, in order to achieve high expected return when evaluating that policy. The data collected for HopperController was taken by training a three layer neural network policy with 64 hidden units and 5126 total weights on the Hopper-v2 MuJoCo task using Proximal Policy Optimization Schulman et al. (2017). Specifically, we use the default parameters for PPO provided in stable baselines Hill et al. (2018). The dataset we provide with this benchmark has 3200 unique weights. In order to collect this many, we run 32 experimental trials of PPO, where we train for one million steps, and save the weights of the policy every 10,000 environment steps. The policy weights are represented originally as a list of tensors. We first traverse this list and flatten each of the tensors, and we then concatenate each of these flattened tensors into a single training example $x_{\text{Hopper}} \in \mathcal{R}^{5126}$. The result is an optimization problem over neural network weights. After collecting these weights, we perform no additional pre-processing steps. In order to collect scores we perform a single rollout for each $x$ using the Hopper-v2 MuJoCo environment. The horizon length for training and evaluation is limited to 1000 simulation time steps.

### A.6   ANTMORPHOLOGY-V0 & DKITTYMORPHOLOGY-V0

Both morphology tasks are collected by us, and share methodology. The goal of these tasks is to design the morphology of a quadrupedal robot—an ant or a D'Kitty—such that the agent is able to crawl quickly in a particular direction. In order to collect data for this environment, we create variants of the MuJoCo Ant and the ROBEL D'Kitty agents that have parametric morphologies. The goal is to determine a mapping from the morphology of the agent to the average return that an agent trained for a particular intended morphology achieves. We implement this by pre-training a neural network policy using SAC Haarnoja et al. (2018). For both the Ant and the D'Kitty, we train agents for up to three million environments steps, and a maximum episode length of 1000, with all other settings as default. These agents are pre-trained for a fixed gold-standard morphology—the default morphology of the Ant and D'Kitty respectively. Each morphology task consists of samples obtained by adding Gaussian noise with standard deviation 0.02 for Ant and 0.01 for DKitty times the design-space range to the gold-standard morphology. We label each sampled morphology by averaging the return of 16 rollouts of length 100 of an agent with that morphology.

## B   ORACLE FUNCTIONS

We detail oracle functions for evaluating ground truth scores for each of the tasks in design-bench. A common thread among these is that the oracle, if trained, is fit to a larger static dataset containing higher performing designs than observed by a downstream MBO algorithm.

### B.1   GFP-V0

GFP-v0 uses the oracle function from Brookes et al. (2019). This oracle is a Gaussian Process regression model with a protein-specific kernel proposed by Shen et al. (2014). The Gaussian Process is fit to a larger dataset than the static dataset packaged with GFP-v0, making it possible to sample a protein design that achieve a higher score than any other protein seen during training. The oracle score for GFP-v0 is implemented as the mean prediction of this Gaussian Process.

## B.2 GFP-v1

GFP-v1 uses an oracle from Rao et al. (2019). This oracle is a Transformer Vaswani et al. (2017) model with 12 layers based on BERT Devlin et al. (2019), and pretrained using masked language modelling and the Pfam database El-Gebali et al. (2019). Instructions for obtained this oracle can be found in the TAPE framework at `https://github.com/songlab-cal/tape`.

## B.3 MOLECULEACTIVITY-V0

Following the procedure set by Martin et al. (2019), the oracle function we use for MoleculeActivity-v0 is a random forest regression model. In particular, we use the RandomForestRegressor provided in scikit-learn, using identical hyperparameters to the random forest regression model used in Martin et al. (2019). The random forest is trained on the entire task dataset. In practice, samples that score at most the 80th percentile are observed by an MBO algorithm, which allows for sampling *unobserved* points that score higher than the highest training point.

## B.4 SUPERCONDUCTOR-V0

The Superconductor-v0 oracle function is also a random forest regression model. The model we use it the model described by Hamidieh (2018). We borrow the hyperparameters described by them, and we use the RandomForestRegressor provided in scikit-learn. Similar to the setup for the previous two tasks, this oracle is trained on the entire static dataset, and the task is instantiated with a split percentile. Samples scoring at most in the 80th percentile are observed by an MBO algorithm, which allows for sampling *unobserved* points that score in the unobserved top 20 percent.

## B.5 HOPPERCONTROLLER-V0

Unlike the previously described tasks, HopperController-v0 and the remaining tasks implement an exact oracle function. For HopperController-v0 the oracle takes the form of a single rollout using the Hopper-v2 MuJoCo environment. The designs for HopperController-v0 are neural network weights, and during evaluation, a policy with those weights is instantiated—in this case that policy is a three layer neural network with 11 input units, two layers with 64 hidden units, and a final layer with 3 output units. The intermediate activations between layers are hyperbolic tangents. After building a policy, the Hopper-v2 environment is reset and the reward for 1000 time-steps is summed. That summed reward constitutes the score returned by the HopperController-v0 oracle. The limit of performance is the maximum return that an agent can achieve in Hopper-v2 over 1000 steps.

## B.6 ANTMORPHOLOGY-V0 & DKITTYMORPHOLOGY-V0

The final two tasks in design-bench use an exact oracle function, using the MuJoCo simulator. For both morphology tasks, the simulator performs 16 rollouts and averages the sum of rewards attained over them. Each task is accompanied by a pre-trained neural network policy. To perform evaluation, a morphology is passed to the Ant or D'Kitty MuJoCo environments respectively, and a dynamic-morphology agent is initialized inside these environments. These environments are very sensitive to small morphological changes, and exhibit a high degree of stochasticity as a result. To compensate for the increased stochasticity, we average returns over 16 rollouts.

## C ADDITIONAL EXPERIMENTAL RESULTS

In this section, we present the 50th percentile performance of the trials presented in main body. Similar to the 100th percentile performance reported in the main text, scores are calculated by evaluating solutions to each task found by an optimization method, subtracting the minimum score present in the corresponding task dataset, and dividing by the range of scores present in the corresponding task dataset. The result is a performance of greater than one if optimization converges to a solution with a higher score that the best observed point in the corresponding task dataset.

| Algorithm Name | GFP-v0 | GFP-v1 | MoleculeActivity-v0 | Superconductor-v0 |
|---|---|---|---|---|
| **Autofocus** | $1.159 \pm 0.072$ | $1.573 \pm 0.003$ | $0.699 \pm 0.059$ | $0.427 \pm 0.101$ |
| **CbAS** | $1.291 \pm 0.045$ | $1.575 \pm 0.002$ | $0.692 \pm 0.046$ | $0.435 \pm 0.098$ |
| **MINs** | $0.957 \pm 0.047$ | $1.535 \pm 0.029$ | $0.717 \pm 0.029$ | $0.504 \pm 0.142$ |
| **Gradient Ascent** | $0.356 \pm 0.000$ | $-1.780 \pm 0.007$ | $0.941 \pm 0.070$ | $0.731 \pm 0.068$ |
| **Grad. Min** | $0.410 \pm 0.131$ | $-1.772 \pm 0.011$ | $0.938 \pm 0.033$ | $0.693 \pm 0.044$ |
| **Grad. Mean** | $0.395 \pm 0.049$ | $-1.584 \pm 0.716$ | $0.945 \pm 0.046$ | $0.658 \pm 0.036$ |
| **REINFORCE** | $0.634 \pm 0.003$ | $-1.476 \pm 0.072$ | $0.694 \pm 0.014$ | $0.694 \pm 0.014$ |
| **BO-qEI** | $0.999 \pm 0.000$ | $1.532 \pm 0.000$ | $0.791 \pm 0.008$ | $0.991 \pm 0.000$ |
| **CMA-ES** | $0.356 \pm 0.000$ | $-1.779 \pm 0.001$ | $0.793 \pm 0.013$ | $0.725 \pm 0.020$ |

Table 4: **50th percentile** evaluations for baselines on every task. Results are averaged over 16 trials, and the $\pm$ indicates the standard deviation of the reported performance. For a description of the score normalization methodology, refer to the caption of Table 2.

| Algorithm Name | HopperController-v0 | AntMorphology-v0 | DKittyMorphology-v0 |
|---|---|---|---|
| **Autofocus** | $0.085 \pm 0.014$ | $1.284 \pm 0.249$ | $0.919 \pm 0.044$ |
| **CbAS** | $0.097 \pm 0.018$ | $1.660 \pm 0.069$ | $0.938 \pm 0.018$ |
| **MINs** | $0.382 \pm 0.221$ | $1.317 \pm 0.123$ | $0.979 \pm 0.067$ |
| **Gradient Ascent** | $0.136 \pm 0.054$ | $1.871 \pm 0.050$ | $1.193 \pm 0.031$ |
| **Grad. Min** | $0.294 \pm 0.042$ | *running* | *running* |
| **Grad. Mean** | $0.266 \pm 0.059$ | *running* | *running* |
| **REINFORCE** | $0.031 \pm 0.002$ | *running* | *running* |
| **BO-qEI** | $0.296 \pm 0.013$ | *running* | *running* |
| **CMA-ES** | $0.035 \pm 0.003$ | *running* | *running* |

Table 5: **50th percentile** evaluations for baselines on every task. Results are averaged over 16 trials, and the $\pm$ indicates the standard deviation of the reported performance. For a description of the score normalization methodology, refer to the caption of Table 2.

## D UNNORMALIZED PERFORMANCE

In this section, we present the unnormalized raw performance of every experiment previously reported in the paper. These tables are displayed over the next several pages.

| | GFP-v0 | GFP-v1 | MoleculeActivity-v0 | Superconductor-v0 |
|---|---|---|---|---|
| **Autofocus** | $3.365 \pm 0.023$ | $3.805 \pm 0.001$ | $6.345 \pm 0.141$ | $77.07 \pm 11.11$ |
| **CbAS** | $3.408 \pm 0.029$ | $3.805 \pm 0.0$ | $6.301 \pm 0.131$ | $72.17 \pm 8.652$ |
| **MINs** | $3.315 \pm 0.033$ | $3.805 \pm 0.0$ | $6.508 \pm 0.236$ | $80.23 \pm 10.67$ |
| **Gradient Ascent** | $2.894 \pm 0.001$ | $1.461 \pm 0.007$ | $6.636 \pm 0.066$ | $89.64 \pm 9.201$ |
| **Grad. Min** | $2.928 \pm 0.074$ | $1.534 \pm 0.162$ | $6.541 \pm 0.075$ | $83.5 \pm 8.934$ |
| **Grad. Mean** | $2.924 \pm 0.039$ | $1.802 \pm 0.711$ | $6.525 \pm 0.116$ | $88.01 \pm 10.432$ |
| **REINFORCE** | $3.252 \pm 0.049$ | $3.803 \pm 0.002$ | $6.465 \pm 0.075$ | $86.58 \pm 4.27$ |
| **BO-qEI** | $3.152 \pm 0.0$ | $3.805 \pm 0.0$ | $7.147 \pm 0.164$ | $89.73 \pm 0.0$ |
| **CMA-ES** | $2.894 \pm 0.0$ | $1.474 \pm 0.004$ | $6.774 \pm 0.096$ | $98.59 \pm 5.271$ |

Table 6: **100th percentile** evaluations for baselines on every task. Results are averaged over 16 trials, and the $\pm$ indicates the standard deviation of the reported performance. This table corresponds to the unnormalized performance originally in Table 2.

## E NORMALIZATION OF GRADIENT ASCENT

The goal of this section is to discuss the normalization method that is used in our Gradient Ascent baseline. For continuous design-space tasks, we normalize both the designs, and the scores to have unit Gaussian statistics. For discrete design-space tasks, we normalize only the scores to have unit Gaussian statistics. This is a necessary part of the optimization workflow because scores vary by several orders of magnitude in the dataset, as low as 2.9 for GFP-v0 and as high as 1400 for HopperController-v0. As we have shown empirically in Figure 7 The specific normalization

| | HopperController-v0 | AntMorphology-v0 | DKittyMorphology-v0 |
|---|---|---|---|
| **Autofocus** | $443.8 \pm 142.9$ | $386.9 \pm 10.58$ | $376.3 \pm 47.47$ |
| **CbAS** | $547.1 \pm 423.9$ | $393.0 \pm 3.75$ | $369.1 \pm 60.65$ |
| **MINs** | $746.1 \pm 636.8$ | $388.5 \pm 9.085$ | $352.9 \pm 38.65$ |
| **Gradient Ascent** | $1050.8 \pm 284.5$ | $399.9 \pm 4.941$ | $390.7 \pm 49.24$ |
| **Grad. Min** | $1952.2 \pm 445.773$ | *running* | *running* |
| **Grad. Mean** | $1877.1 \pm 704.203$ | *running* | *running* |
| **REINFORCE** | $553.0 \pm 295.995$ | *running* | *running* |
| **BO-qEI** | $759.4 \pm 142.341$ | *running* | *running* |
| **CMA-ES** | $676.5 \pm 344.52$ | *running* | *running* |

Table 7: **100th percentile** evaluations for baselines on every task. Results are averaged over 16 trials, and the $\pm$ indicates the standard deviation of the reported performance. This table corresponds to the unnormalized performance originally in Table 3.

| | GFP-v0 | GFP-v1 | MoleculeActivity-v0 | Superconductor-v0 |
|---|---|---|---|---|
| **Autofocus** | $3.216 \pm 0.029$ | $3.8 \pm 0.002$ | $5.759 \pm 0.158$ | $31.57 \pm 7.457$ |
| **CbAS** | $3.269 \pm 0.018$ | $3.802 \pm 0.002$ | $5.742 \pm 0.123$ | $32.21 \pm 7.255$ |
| **MINs** | $3.135 \pm 0.019$ | $3.774 \pm 0.02$ | $5.806 \pm 0.078$ | $37.32 \pm 10.5$ |
| **Gradient Ascent** | $2.894 \pm 0.0$ | $1.457 \pm 0.005$ | $6.401 \pm 0.186$ | $54.06 \pm 5.06$ |
| **Grad. Min** | $2.916 \pm 0.053$ | $1.462 \pm 0.008$ | $6.393 \pm 0.087$ | $51.292 \pm 3.285$ |
| **Grad. Mean** | $2.91 \pm 0.02$ | $1.594 \pm 0.501$ | $6.412 \pm 0.123$ | $48.716 \pm 2.637$ |
| **REINFORCE** | $3.006 \pm 0.001$ | $1.67 \pm 0.05$ | $5.747 \pm 0.038$ | $51.358 \pm 1.059$ |
| **BO-qEI** | $3.152 \pm 0.0$ | $3.772 \pm 0.0$ | $6.004 \pm 0.02$ | $73.361 \pm 0.0$ |
| **CMA-ES** | $2.894 \pm 0.0$ | $1.458 \pm 0.0$ | $6.01 \pm 0.033$ | $53.641 \pm 1.452$ |

Table 8: **50th percentile** evaluations for baselines on every task. Results are averaged over 16 trials, and the $\pm$ indicates the standard deviation of the reported performance. This table corresponds to the unnormalized performance originally in Table 4.

equation for continuous-valued designs is given below.

$$\tilde{\mathbf{x}}_{i,j} = \frac{\mathbf{x}_{i,j} - \mu(\mathbf{x}_{,j})}{\sigma(\mathbf{x}_{,j})} \quad : \quad \mathbf{x} \in \mathbb{R}^{N \times D} \tag{3}$$

We also normalize the objective values in a similar fashion to have unit Gaussian statistics. The result in a new set of designs $\tilde{\mathbf{x}}$ and objective values $\tilde{y}$ that is optimized over.

$$\tilde{y}_{i,j} = \frac{y_{i,j} - \mu(y_{,j})}{\sigma(y_{,j})} \quad : \quad y \in \mathbb{R}^{N \times 1} \tag{4}$$

The gradient ascent procedure is performed in the space of these normalized designs. Suppose $T$ steps of gradient ascent have been taken, and a final normalized solution $\tilde{\mathbf{x}}_T^*$ is found. This solution is de-normalized using the following transformation.

$$(\mathbf{x}_T^*)_{ij} = (\tilde{\mathbf{x}}_T^*)_{ij} \cdot \sigma(\mathbf{x}_{,j}) + \mu(\mathbf{x}_{,j}) \tag{5}$$

This normalization strategy is heavily inspired by data whitening, which is known to reduce the variance of machine learning algorithms that learn discriminative mappings on that data. The learned model of the objective function is one such discriminative model, and normalization likely improves the consistency of Gradient Ascent across independent experimental trials.

## F  HYPERPARAMETER SELECTION WORKFLOW

Hyperparameter tuning under a restricted computational budget is emerging as an import research domain in optimization Sivaprasad et al. (2019); Dodge et al. (2019); Jordan et al. (2020). Care must be taken when tuning each of the prescribed algorithms so that only offline information about the task is used for hyperparameter selection. Formally, this means that the hyperparameters, $\mathcal{H}$, are conditionally independent of the particular value of the performance metric $\mathcal{M}$, given the offline task dataset $\mathcal{D}$. Examples of hyperparameter selection strategies that violate this requirement might, for example, perform a grid search over $\mathcal{H}$ and take the set that maximizes the performance metric,

| | HopperController-v0 | AntMorphology-v0 | DKittyMorphology-v0 |
|---|---|---|---|
| **Autofocus** | $116.4 \pm 18.66$ | $176.7 \pm 59.94$ | $199.3 \pm 8.909$ |
| **CbAS** | $132.5 \pm 23.88$ | $267.3 \pm 16.55$ | $203.2 \pm 3.58$ |
| **MINs** | $520.4 \pm 301.5$ | $184.8 \pm 29.52$ | $211.6 \pm 13.67$ |
| **Gradient Ascent** | $185.0 \pm 72.88$ | $318.0 \pm 12.05$ | $255.3 \pm 6.379$ |
| **Grad. Min** | $400.091 \pm 57.042$ | *running* | *running* |
| **Grad. Mean** | $362.547 \pm 80.091$ | *running* | *running* |
| **REINFORCE** | $42.671 \pm 3.067$ | *running* | *running* |
| **BO-qEI** | $403.363 \pm 17.854$ | *running* | *running* |
| **CMA-ES** | $47.392 \pm 4.416$ | *running* | *running* |

Table 9: **50th percentile** evaluations for baselines on every task. Results are averaged over 16 trials, and the $\pm$ indicates the standard deviation of the reported performance. This table corresponds to the unnormalized performance originally in Table 5.

but this is not offline. An example of a tuning strategy that is fully offline is tuning the parameters of a learned model such that is is a good fit for the task dataset $\mathcal{D}$. One can choose $\mathcal{H}$ that minimizes a validation loss, such as negative log likelihood. A detailed record of hyperparameters can be found in the experiment scripts located at the website for design-bench: `https://sites.google.com/view/design-bench/`.

We now present specific guidelines for hyperparameter selection (i.e. *workflow*) for some of the methods evaluated in our benchmark. These principles are general principles that can be used to tune the hyperparameters of these methods on a new task in a completely offline fashion.

### F.1   STRATEGY FOR AUTOFOCUSED ORACLES

The main tunable components of Autofocused methods (Fannjiang & Listgarten, 2020) are the learned objective function, and the generative model fit to the data distribution. When training the learned objective function, tracking a validation performance metric like rank correlation is helpful to ensure that the resulting learned model is able to generalize beyond its training dataset. This tracking is especially important for Autofocused methods because re-fitting the learned objective model during importance can lead to divergence if the importance weights generated by Autofocusing are very large or very small in magnitude. The algorithm is tuned well if, for example, the validation rank correlation stays above a positive threshold, such as a threshold of 0.9.

The second component of Autofocused methods is the fit of the generative model used for sampling designs. The algorithm has the best chance of success if the generative model can generalize beyond the dataset in which it was trained. This can be monitored by holding out a validation set and tracking a metric such as negative log likelihood on this held-out set. In the case when the generative model is not an exact likelihood-based generative model—for example, a VAE—other validation metrics can be used that measure the fit of the generative model on a validation set. The generative model is especially impacted by the importance sampling procedure used by Estimation of Distribution Algorithms (EDAs), and tracking the effective sample size of the importance weights can help diagnose when the generative model is failing to generalize to a validation set.

### F.2   STRATEGY FOR CBAS

The main tunable components of CbAS methods (Brookes et al., 2019) are the learned objective function, and the generative model fit to the data distribution. While the learned objective function is not affected by the importance sampling weights generated by CbAS, the same tuning strategy described in section F.1 that focuses on generalization to a validation set is effective. Generative model tuning can also follow an identical strategy to that described in section F.1, which focuses on the ability for the generative model to represent samples outside of its training set. In the case of a $\beta$-VAE, which is used with CbAS in this work, the main parameter for controlling this generalization ability is the $\beta$ parameter. We found that $\beta$ is task specific, and must be found in order for the CbAS optimizer using $\beta$-VAE to generate samples that are in the same distribution as its validation set. This value can be tuned in practice using a validation metric like that in section F.1.

### F.3 Strategy For MINs

The main tunable components of MINs (Kumar & Levine, 2019) are the learned objective function, and the generative model fit to the data distribution. The learned objective function is typically trained using a maximum likelihood objective, and the validation log-likelihood (or regression error) can be directly tracked. The learned objective function should train until a minimum validation loss is reached, which ensured that the model will generalize as well as possible beyond its training set. Since only the static task dataset is used for this—it may be split into a training and validation set—this tuning strategy is fully offline.

The generative model for MINs is an inverse mapping $\mathbf{x} = f^{-1}(y, \mathbf{z})$, conditioned on the objective value $y$. Training conditional generative models is considerable less stable than unconditional generative models, so in addition to monitoring the fit of a validation set recommended in section F.1, it is also necessary to track the extent of the dependence of the generative model's predictions on the objective value $y$. This can be evaluated in practice by comparing the distribution of $x$ from the conditional generative model $p(\mathbf{x}|y)$ to an unconditional generative model $p(\mathbf{x})$ with an identical initialization, or by comparing if $p(\mathbf{x}|y)$ is independent of $y$ by querying the inverse model for different values of $y$ and visualizing the similarity in the predictions of $\mathbf{x}$. One metric for more formally studying the extent of the dependence of $\mathbf{x}$ on $\mathbf{z}$ is the mutual information $I(\mathbf{x}; \mathbf{z})$. The conditional generative model has an appropriate fit if for some positive threshold $c$ we have that $I(\mathbf{x}; \mathbf{z}) > c$.

### F.4 Strategy For Gradient Ascent

The main tunable components of Gradient Ascent MBO methods are the learned objective function, and the parameters for gradient ascent. The learned objective function is typically trained using a maximum likelihood objective, and the methodology for obtaining a high-performing learned objective function is identical to that in section F.3. The second aspect of gradient ascent MBO algorithms are the parameters of the gradient-based optimizer for the designs—such as its learning rate, and the number of gradient steps it performs. The learning rate should be small enough that the gradient steps taken increase the prediction of the learned objective function—if the learning rate is too large, gradient steps may not follow the path of steepest ascent of the objective function. The number of gradient steps is more difficult to tune. The strategy we used is a fixed number of steps, and an offline model-selection criterion to select this parameter is future work.

### F.5 Strategy For REINFORCE

The main tunable components of REINFORCE-based MBO methods are the learned objective function, and the parameters for the policy gradient estimator. The learned objective function is typically trained using a maximum likelihood objective, and the methodology for obtaining a high-performing learned objective function is identical to that in section F.3. The remaining parameters to tune are specific to REINFORCE. The distribution of the policy should be carefully selected to be able to model the distribution of designs. For continuous MBO tasks, a Gaussian distribution is appropriate, and for discrete MBO tasks, a categorical distribution is appropriate. In addition, the learning rate, and optimizer should be selected so that policy updates improve the model-predicted score. If the stability of the gradient estimator is suffering, due to high-variance updates, a density ratio threshold like in PPO (Schulman et al., 2017) can be applied, and a baseline can be subtracted.

### F.6 Strategy For Bayesian Optimization

The main tunable components of Bayesian Optimization MBO methods (Balandat et al., 2019) are the learned objective function, and the parameters for the bayesian optimization loop. The learned objective function is typically trained using a maximum likelihood objective, and the methodology for obtaining a high-performing learned objective function is identical to that in section F.3. For a detailed review of the strengths and weaknesses of various Bayesian Optimization strategies and their hyperparameters, we refer the reader to the BoTorch documentation, available at the BoTorch website `https://botorch.org/docs/overview`. In this work we employ a Gaussian Process as the model, and the quasi-Monte Carlo Expected Improvement acquisition function, which has the advantage of scaling up to our high-dimensional optimization problems.

## F.7 STRATEGY FOR COVARIANCE MATRIX ADAPTATION (CMA-ES)

The main tunable components of Covariance Matrix Adaptation MBO methods are the learned objective function, and the parameters for the evolution strategy. The learned objective function is typically trained using a maximum likelihood objective, and the methodology for obtaining a high-performing learned objective function is identical to that in Subsection F.3. For a detailed review of the strengths and weaknesses of various Bayesian Optimization strategies and their hyperparameters, we refer the reader to an open-source implementation of CMA-ES and its corresponding documentation https://github.com/CMA-ES/pycma. In this work we employ the default settings for CMA-ES reported in this open source implementation, with $\sigma = 0.5$.

## G PERFORMANCE REPORTING CONFIDENCE INTERVALS

This section contains additional results for each baseline algorithm evaluated on every task, reporting confidence intervals rather than the standard deviation of the performance. We calculate the 90th, 95th, and 99th confidence interval for each method, and report that intervals using $\pm$ notation next to the sample mean performance. Results are calculated using 16 trials per method.

| | GFP-v0 | GFP-v1 | MoleculeActivity-v0 | Superconductor-v0 |
|---|---|---|---|---|
| **Autofocus** | 1.531±0.031 | 1.58±0.0 | 0.919±0.029 | 1.041±0.083 |
| **CbAS** | 1.637±0.039 | 1.58±0.0 | 0.903±0.027 | 0.975±0.064 |
| **MINs** | 1.407±0.046 | 1.58±0.0 | 0.981±0.049 | 1.084±0.079 |
| **Gradient Ascent** | 0.356±0.0 | -1.775±0.005 | 1.029±0.034 | 1.215±0.07 |
| **Grad. Min** | 0.44±0.102 | -1.67±0.127 | 0.993±0.015 | 1.128±0.066 |
| **Grad. Mean** | 0.43±0.053 | -1.286±0.56 | 0.987±0.024 | 1.189±0.078 |
| **REINFORCE** | 1.248±0.0 | 1.576±0.005 | 0.965±0.0 | 1.17±0.0 |
| **BO-QEI** | 1.0±0.0 | 1.579±0.0 | 1.221±0.0 | 1.213±0.0 |
| **CMA-ES** | 0.356±0.0 | -1.755±0.018 | 1.081±0.0 | 1.332±0.0 |

Table 10: **100th percentile** evaluations for baselines on every task. Results are averaged over 16 trials, and the $\pm$ indicates the **95th confidence interval** of the reported performance.

| | HopperController-v0 | AntMorphology-v0 | DKittyMorphology-v0 |
|---|---|---|---|
| **Autofocus** | 0.326±0.058 | 2.217±0.074 | 1.787±0.428 |
| **CbAS** | 0.402±0.171 | 2.183±0.029 | 1.752±0.547 |
| **MINs** | 0.548±0.257 | 2.164±0.021 | 1.672±0.104 |
| **Gradient Ascent** | 0.753±0.1 | 2.215±0.04 | 1.668±0.322 |
| **Grad. Min** | 1.434±0.18 | *running* | *running* |
| **Grad. Mean** | 1.379±0.285 | *running* | *running* |
| **REINFORCE** | 0.406±0.0 | *running* | *running* |
| **BO-QEI** | 0.558±0.0 | *running* | *running* |
| **CMA-ES** | 0.497±0.0 | *running* | *running* |

Table 11: **100th percentile** evaluations for baselines on every task. Results are averaged over 16 trials, and the $\pm$ indicates the **95th confidence interval** of the reported performance.

| | GFP-v0 | GFP-v1 | MoleculeActivity-v0 | Superconductor-v0 |
|---|---|---|---|---|
| **Autofocus** | 1.16±0.041 | 1.573±0.002 | 0.699±0.033 | 0.427±0.055 |
| **CbAS** | 1.292±0.025 | 1.575±0.001 | 0.693±0.025 | 0.435±0.054 |
| **MINs** | 0.958±0.026 | 1.535±0.016 | 0.717±0.016 | 0.504±0.078 |
| **Gradient Ascent** | 0.356±0.0 | -1.78±0.004 | 0.95±0.032 | 0.731±0.038 |
| **Grad. Min** | 0.41±0.072 | -1.772±0.006 | 0.938±0.018 | 0.693±0.024 |
| **Grad. Mean** | 0.395±0.027 | -1.584±0.394 | 0.945±0.025 | 0.658±0.02 |
| **REINFORCE** | 0.634±0.0 | -1.476±0.132 | 0.694±0.0 | 0.694±0.0 |
| **BO-QEI** | 0.999±0.0 | 1.532±0.0 | 0.791±0.0 | 0.991±0.0 |
| **CMA-ES** | 0.356±0.0 | -1.779±0.002 | 0.793±0.0 | 0.725±0.0 |

Table 12: **50th percentile** evaluations for baselines on every task. Results are averaged over 16 trials, and the $\pm$ indicates the **95th confidence interval** of the reported performance.

|  | HopperController-v0 | AntMorphology-v0 | DKittyMorphology-v0 |
|---|---|---|---|
| **Autofocus** | 0.086±0.008 | 1.284±0.458 | 0.919±0.08 |
| **CbAS** | 0.097±0.01 | 1.66±0.126 | 0.938±0.032 |
| **MINs** | 0.382±0.122 | 1.317±0.068 | 0.979±0.037 |
| **Gradient Ascent** | 0.136±0.029 | 1.871±0.045 | 1.156±0.117 |
| **Grad. Min** | 0.294±0.023 | *running* | *running* |
| **Grad. Mean** | 0.266±0.032 | *running* | *running* |
| **REINFORCE** | 0.031±0.0 | *running* | *running* |
| **BO-QEI** | 0.296±0.0 | *running* | *running* |
| **CMA-ES** | 0.035±0.0 | *running* | *running* |

Table 13: **50th percentile** evaluations for baselines on every task. Results are averaged over 16 trials, and the ± indicates the **95th confidence interval** of the reported performance.

|  | GFP-v0 | GFP-v1 | MoleculeActivity-v0 | Superconductor-v0 |
|---|---|---|---|---|
| **Autofocus** | 1.531±0.026 | 1.58±0.0 | 0.919±0.024 | 1.041±0.068 |
| **CbAS** | 1.637±0.032 | 1.58±0.0 | 0.903±0.022 | 0.975±0.053 |
| **MINs** | 1.407±0.038 | 1.58±0.0 | 0.981±0.04 | 1.084±0.065 |
| **Gradient Ascent** | 0.356±0.0 | -1.775±0.004 | 1.029±0.026 | 1.215±0.058 |
| **Grad. Min** | 0.44±0.084 | -1.67±0.105 | 0.993±0.013 | 1.128±0.055 |
| **Grad. Mean** | 0.43±0.044 | -1.286±0.46 | 0.987±0.02 | 1.189±0.064 |
| **REINFORCE** | 1.248±0.121 | 1.576±0.004 | 0.965±0.028 | 1.17±0.058 |
| **BO-QEI** | 1.0±0.0 | 1.579±0.0 | 1.221±0.062 | 1.213±0.0 |
| **CMA-ES** | 0.356±0.0 | -1.755±0.012 | 1.081±0.036 | 1.332±0.071 |

Table 14: **100th percentile** evaluations for baselines on every task. Results are averaged over 16 trials, and the ± indicates the **90th confidence interval** of the reported performance.

|  | HopperController-v0 | AntMorphology-v0 | DKittyMorphology-v0 |
|---|---|---|---|
| **Autofocus** | 0.326±0.047 | 2.217±0.054 | 1.787±0.316 |
| **CbAS** | 0.402±0.141 | 2.183±0.021 | 1.752±0.404 |
| **MINs** | 0.548±0.212 | 2.164±0.017 | 1.672±0.086 |
| **Gradient Ascent** | 0.753±0.082 | 2.215±0.032 | 1.668±0.238 |
| **Grad. Min** | 1.434±0.148 | *running* | *running* |
| **Grad. Mean** | 1.379±0.234 | *running* | *running* |
| **REINFORCE** | 0.406±0.217 | *running* | *running* |
| **BO-QEI** | 0.558±0.105 | *running* | *running* |
| **CMA-ES** | 0.497±0.253 | *running* | *running* |

Table 15: **100th percentile** evaluations for baselines on every task. Results are averaged over 16 trials, and the ± indicates the **90th confidence interval** of the reported performance.

|  | GFP-v0 | GFP-v1 | MoleculeActivity-v0 | Superconductor-v0 |
|---|---|---|---|---|
| **Autofocus** | 1.16±0.033 | 1.573±0.001 | 0.699±0.027 | 0.427±0.046 |
| **CbAS** | 1.292±0.021 | 1.575±0.001 | 0.693±0.021 | 0.435±0.044 |
| **MINs** | 0.958±0.021 | 1.535±0.013 | 0.717±0.013 | 0.504±0.064 |
| **Gradient Ascent** | 0.356±0.0 | -1.78±0.003 | 0.95±0.025 | 0.731±0.031 |
| **Grad. Min** | 0.41±0.059 | -1.772±0.005 | 0.938±0.015 | 0.693±0.02 |
| **Grad. Mean** | 0.395±0.022 | -1.584±0.324 | 0.945±0.021 | 0.658±0.016 |
| **REINFORCE** | 0.634±0.003 | -1.476±0.098 | 0.694±0.014 | 0.694±0.014 |
| **BO-QEI** | 0.999±0.0 | 1.532±0.0 | 0.791±0.008 | 0.991±0.0 |
| **CMA-ES** | 0.356±0.0 | -1.779±0.001 | 0.793±0.013 | 0.725±0.02 |

Table 16: **50th percentile** evaluations for baselines on every task. Results are averaged over 16 trials, and the ± indicates the **90th confidence interval** of the reported performance.

|  | HopperController-v0 | AntMorphology-v0 | DKittyMorphology-v0 |
|---|---|---|---|
| **Autofocus** | 0.086±0.006 | 1.284±0.339 | 0.919±0.059 |
| **CbAS** | 0.097±0.008 | 1.66±0.094 | 0.938±0.024 |
| **MINs** | 0.382±0.1 | 1.317±0.056 | 0.979±0.03 |
| **Gradient Ascent** | 0.136±0.024 | 1.871±0.036 | 1.156±0.087 |
| **Grad. Min** | 0.294±0.019 | *running* | *running* |
| **Grad. Mean** | 0.266±0.027 | *running* | *running* |
| **REINFORCE** | 0.031±0.002 | *running* | *running* |
| **BO-QEI** | 0.296±0.013 | *running* | *running* |
| **CMA-ES** | 0.035±0.003 | *running* | *running* |

Table 17: **50th percentile** evaluations for baselines on every task. Results are averaged over 16 trials, and the ± indicates the **90th confidence interval** of the reported performance.

|  | GFP-v0 | GFP-v1 | MoleculeActivity-v0 | Superconductor-v0 |
|---|---|---|---|---|
| **Autofocus** | 1.531±0.043 | 1.58±0.001 | 0.919±0.04 | 1.041±0.114 |
| **CbAS** | 1.637±0.054 | 1.58±0.001 | 0.903±0.038 | 0.975±0.089 |
| **MINs** | 1.407±0.063 | 1.58±0.0 | 0.981±0.068 | 1.084±0.11 |
| **Gradient Ascent** | 0.356±0.0 | -1.775±0.007 | 1.029±0.057 | 1.215±0.097 |
| **Grad. Min** | 0.44±0.14 | -1.67±0.176 | 0.993±0.021 | 1.128±0.092 |
| **Grad. Mean** | 0.43±0.074 | -1.286±0.774 | 0.987±0.033 | 1.189±0.107 |
| **REINFORCE** | 1.248±0.0 | 1.576±0.01 | 0.965±0.0 | 1.17±0.0 |
| **BO-QEI** | 1.0±0.0 | 1.579±0.0 | 1.221±0.0 | 1.213±0.0 |
| **CMA-ES** | 0.356±0.0 | -1.755±0.04 | 1.081±0.0 | 1.332±0.0 |

Table 18: **100th percentile** evaluations for baselines on every task. Results are averaged over 16 trials, and the ± indicates the **99th confidence interval** of the reported performance.

|  | HopperController-v0 | AntMorphology-v0 | DKittyMorphology-v0 |
|---|---|---|---|
| **Autofocus** | 0.326±0.08 | 2.217±0.135 | 1.787±0.786 |
| **CbAS** | 0.402±0.237 | 2.183±0.053 | 1.752±1.004 |
| **MINs** | 0.548±0.356 | 2.164±0.029 | 1.672±0.144 |
| **Gradient Ascent** | 0.753±0.138 | 2.215±0.059 | 1.668±0.591 |
| **Grad. Min** | 1.434±0.249 | *running* | *running* |
| **Grad. Mean** | 1.379±0.393 | *running* | *running* |
| **REINFORCE** | 0.406±0.0 | *running* | *running* |
| **BO-QEI** | 0.558±0.0 | *running* | *running* |
| **CMA-ES** | 0.497±0.0 | *running* | *running* |

Table 19: **100th percentile** evaluations for baselines on every task. Results are averaged over 16 trials, and the ± indicates the **99th confidence interval** of the reported performance.

|  | GFP-v0 | GFP-v1 | MoleculeActivity-v0 | Superconductor-v0 |
|---|---|---|---|---|
| **Autofocus** | 1.16±0.056 | 1.573±0.002 | 0.699±0.045 | 0.427±0.077 |
| **CbAS** | 1.292±0.035 | 1.575±0.002 | 0.693±0.035 | 0.435±0.075 |
| **MINs** | 0.958±0.036 | 1.535±0.022 | 0.717±0.022 | 0.504±0.108 |
| **Gradient Ascent** | 0.356±0.0 | -1.78±0.005 | 0.95±0.053 | 0.731±0.052 |
| **Grad. Min** | 0.41±0.1 | -1.772±0.008 | 0.938±0.025 | 0.693±0.034 |
| **Grad. Mean** | 0.395±0.037 | -1.584±0.545 | 0.945±0.035 | 0.658±0.027 |
| **REINFORCE** | 0.634±0.0 | -1.476±0.242 | 0.694±0.0 | 0.694±0.0 |
| **BO-QEI** | 0.999±0.0 | 1.532±0.0 | 0.791±0.0 | 0.991±0.0 |
| **CMA-ES** | 0.356±0.0 | -1.779±0.004 | 0.793±0.0 | 0.725±0.0 |

Table 20: **50th percentile** evaluations for baselines on every task. Results are averaged over 16 trials, and the ± indicates the **99th confidence interval** of the reported performance.

|  | HopperController-v0 | AntMorphology-v0 | DKittyMorphology-v0 |
|---|---|---|---|
| **Autofocus** | 0.086±0.01 | 1.284±0.84 | 0.919±0.147 |
| **CbAS** | 0.097±0.013 | 1.66±0.232 | 0.938±0.059 |
| **MINs** | 0.382±0.168 | 1.317±0.093 | 0.979±0.051 |
| **Gradient Ascent** | 0.136±0.041 | 1.871±0.066 | 1.156±0.216 |
| **Grad. Min** | 0.294±0.032 | *running* | *running* |
| **Grad. Mean** | 0.266±0.045 | *running* | *running* |
| **REINFORCE** | 0.031±0.0 | *running* | *running* |
| **BO-QEI** | 0.296±0.0 | *running* | *running* |
| **CMA-ES** | 0.035±0.0 | *running* | *running* |

Table 21: **50th percentile** evaluations for baselines on every task. Results are averaged over 16 trials, and the ± indicates the **99th confidence interval** of the reported performance.

