# OpenReview forum: "Design-Bench: Benchmarks for Data-Driven Offline Model-Based Optimization"
_ICLR.cc/2021/Conference — Reject_

### Official Review · AnonReviewer3 · 2020-10-26
**A good start**

**Rating:** 6
**Confidence:** 4

**Review:**

This paper studies the evaluation of offline black-box optimization algorithms. The community currently lacks a standardized benchmark to compare the performance of methods. This paper presents a new suite of offline model-based optimization tasks and standardized evaluation procedures for the community. The evaluation criterion for the quality of a benchmark is the realism and diversity of the tasks, with special consideration for high-dimensional design space and the objective function's sensitivity. The paper then evaluates several algorithms on the benchmark.

The creation of a useful benchmark is an important and challenging task. It seems that consideration was given to the choice of optimization problems. The result is a diverse set, with some representing real-world design optimization problems. However, there are a few areas in which the benchmark and evaluation should be improved.

I do not recommend this paper for acceptance as there is insufficient support as to why these problems should be considered over others or what challenges these environments present for designing new algorithms. There are also some deficiencies in the evaluation protocol.

For the GFP, Molecule, and superconductor problems, an "expert model" is used as the oracle function. The expert model's use speeds up the evaluation and is undoubtedly the right choice in creating a benchmark, but it introduces bias when evaluating an algorithm. The goal of the paper is to present a benchmark for use in the development of novel algorithms. It is important that if others are to use this benchmark to design new algorithms, they should be aware of the benchmark's biases. A more detailed investigation of these specific optimization tasks and what type of algorithms they favored is warranted.


In the morphology tasks, the data is generated using a policy trained with a given robot morphology and then later evaluated using a different morphology. This seems to bias the optimal choice of morphology to be the one the policy was used during training. Can the authors clarify what is intended to be learned from including these tasks in the benchmark?



The evaluation protocol is lacking in two areas: how hyperparameter tuning is considered and how results are compared.

It is said that no hyperparameter tuning can use the oracle function. This is an obvious necessity, but it remains unclear to what extent algorithms are allowed to perform hyperparameter tuning. This makes it unclear if one algorithm performs better than another due to more hyperparameter tuning. Some recent works (Sivaprasad et al. 2020, Jordan et al. 2020, Dodge et al. 2019) address some of these issues when evaluating algorithms.

In comparing the results, it is not clear how one determines which algorithm performs best. No aggregate measure is given to determine performance. Furthermore, it is unclear how the uncertainty of the results are quantified. These methods have some degree of stochasticity in the performance, be it from the algorithm or choice hyperparameters. It is unclear what sources of uncertainty are being considered in the results and how many trials have been executed. What do the +/- numbers represent in tables 2 and 3, and how are they computed?

Small notes for areas of improvement:

In section 7 study, an ablation study is mentioned for gradient descent, but there is no discussion of how this study was carried out. Gradient descent is known to be sensitive to the scaling and parameterization of the function, which has led to the design of many optimizers that use preconditions on the gradients, e.g., Newton's method, natural gradient, Adagrad, RMSProp, etc. This problem has also been directly considered in finding optimal design points (Box and Draper 2007).

The sensitivity (smoothness) of the objective function to small perturbations of its inputs is an important characteristic to consider. The provided example in Figure 2 claims that the objective function is highly sensitive, but the provided example does not appear to have high sensitivity but is rather discontinuous. Not to say that this is not a challenging problem, just that high sensitivity is perhaps not the correct interpretation for this example.

In the MuJoCo environment tasks, the oracle objective function is evaluated 16 times and averaged to reduce noise. How was this number chosen, and how uncertain are the evaluations? It seems unlikely that it can be assumed that with 16 samples, the sample mean is normally distributed.

Additionally, can the authors clarify why 100 and 1000 timesteps are sufficient for the MuJoCo tasks of interest? As presented, these seem like arbitrary choices.

Box, George EP, and Norman R. Draper. Response surfaces, mixtures, and ridge analyses. Vol. 649. John Wiley & Sons, 2007.

Sivaprasad, P. T., Mai, F., Vogels, T., Jaggi, M., & Fleuret, F. (2020). Optimizer benchmarking needs to account for hyperparameter tuning. In Proceedings of the 37th International Conference on Machine Learning.

Jordan, S. M., Chandak, Y., Cohen, D., Zhang, M., & Thomas, P. S. (2020). Evaluating the Performance of Reinforcement Learning Algorithms. In Proceedings of the 37th International Conference on Machine Learning.

Dodge, J., Gururangan, S., Card, D., Schwartz, R., & Smith, N. A. (2019). Show your work: Improved reporting of experimental results. arXiv preprint arXiv:1909.03004.

---

> ### Author Response · Authors · 2020-11-19
> **Author Response: Added new GFP expert model, hyperparameter tuning and workflow details, details of gradient ascent**
>
> We thank the reviewer for detailed comments on our work. To address the reviewer’s concerns, in the revised paper, we have elaborated on a discussion on hyperparameters (Appendix F.1 - F.7) as well as added a discussion of practical guidelines for tuning hyperparameters (i.e., workflow) for a new algorithm (Appendix F). We have added a new task GFP-v1 that uses a state-of-the-art TAPE model as the expert and we find that the relative performance of different methods is mostly the same with this model (Tables 2 and 3). We have updated a discussion of normalization in gradient descent (Appendix E).
>
> **Unbiased nature of expert models used for evaluation**
> We have added a new version of the GFP task where we utilize the state-of-the-art Transformer model from the TAPE suite (Rao et al. 2019) as the groundtruth objective function for the GFP task (now indicated as GFP-v1 in Table 2 and 3). We observe that even with this evaluation function, distinct from the GP-based oracle (used in GFP-v0), the relative performance of different methods is highly correlated with GFP-v0, and the rank correlation between the performance of the methods compared against these two different ground truth functions is **0.937**. While this does not indicate that the expert models are unbiased, it does suggest that the solutions obtained are robust to the choice of the expert model used for annotating the objective values in the dataset.
>
> **Morphology Tasks**
> The reviewer’s understanding is correct -- the goal of these tasks is to find the morphology such that running a pre-specified policy (unknown to the algorithm) on the produced morphology gives the highest return. The goal is to evaluate the ability of the algorithm to optimize over a space of morphologies where the objective function is a stochastic and complex function of the design space.
>
> **Hyperparameter tuning and workflow**
> We performed hyperparameter tuning to the extent that can be performed completely offline for each method as we discuss in Section 7 and we have now elaborated the discussion in Appendix F. This means that different values of hyperparameters can be selected for each method, provided that all tuning is done completely offline. We have added the references in related work and provide some *”workflow”* guidelines which can be used by researchers to tune future algorithms in Appendix F. We have also cited the papers pointed out by the reviewer pertaining to hyperparameter tuning.
>
> **Performance results, uncertainty in Table 2 and 3**
> The +/- numbers in these Tables denote the standard deviation of the groundtruth score of the optimized inputs over 16 independent training runs. Formally, Let, $\{x^i_1, x^i_2, \cdots, x^i_N\}$ denote the set of $N = 128$ optimized samples generated in the i^th training run. Then, the entries in Tables 2 are given by $mean(\max_{j=1, \cdots, N} f(x)^i_j) \pm std(\max_{j=1, \cdots, N} f(x)^i_j)$ and the results in Appendix C are given by $mean(50^{th}percentile_{j=1, \cdots, N} f(x)^i_j) \pm std(50^{th}percentile_{j=1, \cdots, N} f(x)^i_j)$.
>
> **“Why should these problems be considered and what challenges they present for new algorithms?”**
> We provide a discussion of various task properties that we considered in choosing tasks for our benchmark in Section 5. Through these properties, we identify a set of unique challenges for offline MBO algorithms including high-dimensional design spaces (which is particularly challenging for GP-based or search  algorithms) and highly sensitive objective functions (which is challenging for algorithms that cannot model discontinuous function behavior). We have also added the discussion of a new design factor which pertains to the long-tailed nature of the data distribution, which is challenging for neural network based methods.
>
> As discussed in Section 5, the tasks in our benchmark possess these desired design properties, and as a result these problems were considered to be in our benchmark. As can be seen in our results, these methods do not solve many tasks perfectly, attaining a performance worse than the best in the dataset. These results indicate that our tasks do present sufficient challenges for current offline MBO methods. We are happy to add more problems or more design factors in the benchmark if the reviewer has any suggestions.
>
> **MuJoCo tasks design choices**
> We used 16 independent runs for training since typically prior works in reinforcement learning find it sufficient to use ~10 seeds for MuJoCo tasks. 1000 is the typical choice of rollout horizon for MuJoCo tasks, which is what we used for HopperController. We used 100 for the Morphology tasks, since this number allowed us to retain the task complexity (longer horizon typically gives rise to harder tasks) while still allowing faster evaluation in terms of wall-clock time, since the DKitty simulator is slow to run rollouts in.

---

> > ### Comment · AnonReviewer3 · 2020-11-22
> > **An improved paper**
> >
> > Thanks for the detailed clarification and additions to the paper. Most of my concerns and questions have now been addressed.
> >
> > One small way the paper could still be improved is to use confidence intervals instead of standard deviations in reporting the results. This change will make it clear which algorithm comparisons have sufficient empirical evidence.
> >
> > For the number of seeds on MuJoCo tasks, there is ample evidence that ten seeds are not sufficient for reliable comparisons (Colas et al., 2018, Henderson et al., 2018). There is no set number of trials to ensure that a reliable comparison is made, which is why confidence intervals should be used regardless of sample size.
> >
> > Colas, C., Sigaud, O., & Oudeyer, P. (2018). How Many Random Seeds? Statistical Power Analysis in Deep Reinforcement Learning Experiments. ArXiv, abs/1806.08295.
> >
> > Henderson, P., Islam, R., Bachman, P., Pineau, J., Precup, D., & Meger, D. (2018). Deep Reinforcement Learning that Matters. AAAI.

---

> > > ### Author Response · Authors · 2020-11-23
> > > **Thank you for the follow up! Added Confidence Intervals to Results**
> > >
> > > We thank the reviewer for their prompt follow-up. We are glad that our response addresses most of the reviewer’s concerns and questions.
> > >
> > > To address the reviewer’s concern about mentioning confidence intervals instead of standard deviations, we have now added the results for all tasks in the form of confidence intervals in Tables 2 and 3 in the main paper and elaborate versions for 90th, 95th, and 99th confidence intervals for these tables in Appendix G. These changes are indicated in purple color. We found these intervals tend to have the same relative magnitudes as the standard deviations we originally reported.
> > >
> > > We are happy to resolve any other concerns that the reviewer has. We would appreciate it if the reviewer can kindly update their assessment of the paper in light of this discussion.

---

> > > > ### Comment · AnonReviewer3 · 2020-11-24
> > > > **Updated score**
> > > >
> > > > Thanks for adding these. You should also mention the method of computing the confidence intervals.
> > > >
> > > > I have updated more score to recommend the paper for acceptance.

---

> > > > > ### Author Response · Authors · 2020-11-25
> > > > > **Thank you for the re-evaluation! Added Confidence Interval Calculation Details.**
> > > > >
> > > > > We thank the reviewer for their positive assessment and for increasing their rating.
> > > > > We have added the details about how we calculate confidence intervals to Table 2 in the revised paper.

---

> ### Author Response · Authors · 2020-11-22
> **Discussion**
>
> Dear reviewer,
>
> Please let us know if our response below addresses the concerns raised in your review. We will be happy to clarify these or other concerns more.

---

### Official Review · AnonReviewer2 · 2020-10-28
**Valuable if done well, but needs substantial improvement**

**Rating:** 5
**Confidence:** 4

**Review:**

Summary:
This paper focuses on model-based black box optimization problems in the offline setting. These are settings where access to ground truth is expensive, and instead the optimizer has access only to a trained model of the ground truth based on limited data. While optimizing on this surrogate space, a good optimizer often needs to account for model uncertainty and accuracy degradation. The main aim of the paper is to provide a test bed for algorithms that try to solve this challenge.

Positives:
This is a well-motivated line of work because there is a large interest in these problems across fields. There is indeed a need for better benchmarks and better libraries to make it easy to compare methods. I think the paper is executed cleanly and the it's well-written. I also think the work, when completed, has potential to be very useful.

Areas for improvement:

In my view, the paper has shortcomings in it's design, development and scope. A paper like this is helpful when it can:

 (i) Establish good practices across the board by streamlining workflow, and ensure the interface is used the same way when comparing methods.

(ii) Contribute code that make it easy and fast to use and develop on.

(iii)  Make it easy to collect and report relevant performance statistics the same way across algorithms, helping the field by making it easy to benchmark.

(iv) Includes challenges that are diverse but relevant to the use case of the algorithm.

(v) Synthesize a suite of methods from the literature that are distinct from each other and of interest to the community as strong benchmarks.

I don't think the paper is addressing these aspects sufficiently. Some detailed comments below.

(1) The categorical choice of benchmark tasks is not clearly justified (e.g. Why should anyone care if a protein design algorithm is poor at designing a robot controller?) At the end of the day, these problems share little structure, and "no free lunch" arguments (Wolpert 95) would suggest that there is no one good algorithm for every challenge. The real world settings for these problems don't map well to each other. This paper as designed, could result in follow up work with meaningless comparisons between algorithms that have no business being compared (unless there is a meaningful connection between the challenges). If the authors feel like they can justify this particular set of challenges, I'm open to be convinced. E.g. if the argument is that there will be a "master algorithm" that is just good at designing everything, and this benchmarking set is designed to enable that, I could drop this point, but then other issues will be relevant.

(2) The particular choices within each class of tasks is insufficiently justified. Why is GFP a good design challenge given that the ground truth is also necessarily a trained oracle with likely poor performance outside the training data? Just because some previous papers chose this as the design task doesn't make it a good benchmark. No statistics is provided for how good the GP model for GFP is.  As for other proteins, GB1 for instance is far more completely surveyed than GFP.  There are also better published models of GFP available (e.g. TAPE, UniRep). All of these would of course struggle with out-of-domain samples. Why not use a physical simulator like Rosetta that wouldn't have this issue? I believe when designing a benchmarking suite, these decisions should be considered more carefully, as it would quickly become the test bed for follow up work and a flawed design choice here amplifies in future work.

(3) A body of literature for algorithms that can readily perform MBO has been neglected.  It is trivial to run a regular optimization algorithm with the model (instead of ground truth) and compare the proposed solutions to ground truth.  Quality-Diversity/EDA algorithms (e.g. genetic algorithms,  simulated annealing,  CMA-ES, or even pure CEM (rather than DbAS)...) consistently perform "well" in these high-dimensional optimization settings.  The success of the gradient-based method gives more reason to believe representatives of each of these classical approaches should be included and suggests that the claim that climbing proxy model will necessarily result in bad "ground truth" outcomes is a weak one.

(4) For a benchmarking library like this, there needs to be mature code available for review (not submitted). I've checked the provided website multiple times, and while it is under active development, the code is not accessible. From what I gather the current code interface simply gives access to some data points and a ground truth. This is too little API. A good benchmarking tool would let the user abstract away the modeling part easily, and be able to readily port and run their algorithm against benchmarks, producing the results in the same way. It should also take care of running sanity checks/tests for the user and generating the same plots as those in the paper.

(5) The fact that gradient-based methods outperform other methods presented here is only surprising in the sense that they were not included in the original papers (i.e. why weren't gradient-based methods benchmarked there? not this paper's fault of course).  The authors express  a general conviction where gradient based methods have done very poorly in other attempts for MBO, but provide no references, it would be great to cite relevant references for this claim.

(6) As is, the paper/library only compares CbAS/DbAS with MINs and hill-climbing methods. I think it is not sufficient breadth of methods to make a "benchmarking" suite. As far as I can tell, CbAS/DbAS and MINs are not the best published algorithms in any of the domains suggested, so the authors should justify why they are the algorithms to benchmark against?

For instance, Angermueller et al 2020 ICLR, have an offline RL algorithm that can in principle solve all of these problems. In fact DynaPPO, PPO, and Ensemble-based Bayesian optimization all outperform CbAS in that study. Some sequence design challenges used there seem to be better benchmarks than GFP.  There is substantive work in molecular design on MBO,  but none of the SOTA algorithms are included (e.g the now-classic Gomez-Bombarelli 2016, or  perhaps adaptation of Zhou et al 2019 Scientific Reports). A good rule of thumb in my view is to include the SOTA or well-established algorithm for each task category.

(6) I suggest the authors think carefully about what the evaluation criteria are. While optimization itself is a good metric, other factors, such as providing a way of evaluating diversity of solutions, or sensitivity to dataset size (e.g. by subsampling), are good to consider.

==========

I would like to encourage the authors to continue the pursuit of this work because it is relevant and well-motivated, and has great potential, in my view it is simply not ready. I think this work needs to be reviewed again when it is more mature,  with wider range of algorithms, better justified challenges, a larger set of metrics that can be easily collected, and available code such that the reviewer can vet the benchmarks and code properly. Right now, it's a comparative review of a small set of methods, not a good benchmarking suite.

If done well, it can be a very useful suite that can help researchers develop better algorithms. The danger of accepting it prematurely is that it will be a basis for future work that "game-ify" studies of algorithms against irrelevant/misleading set of benchmarks. That is only damaging to the development of good algorithms and could misguide research. As it stands,  I find the latter risk higher than it's contribution, and hence I believe it should be rejected and reviewed once more of these structural issues are addressed (and code is available to review).


~~~~~~~~~~~~~~

Post review and discussion remarks:
I think the authors have improved the paper significantly during the review period. However, three of my main concerns about the paper remain to a degree that I'm not confident about the paper's value (or risk of misleading followups). (1) That the set of challenges is somewhat arbitrary, some tasks are using "real" ground truths while others are simply running on known trained oracles.  (2) That implementation of strong offline RL benchmark algorithms are missing (because they don't exactly apply across domains) even though they can always be applied in this setting even if "exact" conditions are not applied in every case (just like Gradient Ascent or BO were) (3) That the API needs to offer more for this to be a good benchmarking suite.

I've been most concerned about 2 and 3, and after reading the code, I find that it is still too "bare bones" to be a good package. I looked back at OpenAI gym, and there are several abstractions that they make, including actions, observations, environments, spaces that help the implementer unify how they deal with the complexity underneath.

So far as I can tell, most of what design-bench does is load a csv matrix into an task.x, task.score(x) , and also lets the user access some approximate oracle task.y. This are critical to the process but their abstraction as related to the paper are not clear to me at this time. How is task.y computed, how is the ground truth actually representative of reality. How does optimization depend on the choice of oracle for task.y?

Having read the code, useful elements are in there to make for a good package, I feel like it needs improvements and another review for scientific soundness.

I've updated my score to address the improvements made. The paper scores somewhere between a 4 and a 5 for me.

---

> ### Author Response · Authors · 2020-11-19
> **Author Response (Part 1 of 2):  Summary, Master Algorithm, GP oracle model**
>
> We thank the reviewer for the detailed review and the constructive comments for helping us to build a better benchmark. To address the reviewer’s concerns, we have added a new task to Design-Bench (GFP-v1) that uses the state-of-the-art transformer model from the TAPE suite (Rao et al. 2019) as the groundtruth fluorescence value. We have added several new baseline methods: (1) methods that use ensembles of learned objective models (2) CMA-ES (genetic algorithm) (3) Bayesian Optimization (4) a variant of DynaPPO from Angermueller et al. ICLR 2020. We have also released the code for the baseline algorithms on the anonymous website located at https://sites.google.com/view/design-bench/home#h.bg17kh984oyh. The changes to the updated paper are shown in blue.
>
> **Summary  of some improvements**
>
> - We originally provided code for the benchmark tasks, but have since simplified the process of viewing code for the tasks along with access to ground truth functions via zip files at the anonymous URL https://sites.google.com/view/design-bench .
>
> - We have updated the website to include code for standardized implementation of 9 baseline algorithms as well.
>
> - We have added a discussion on the workflow with this benchmark in Appendix F of the paper.
>
> - And we have now evaluated **5** new methods that include methods using a generative model, methods using gradient-based optimization, methods using evolutionary strategies as well as a Bayesian optimization method.
>
> - Our tasks come from a diverse range of domains and are motivated by a set of principled design factors (Section 5).
>
> We elaborate on the questions posed by the reviewers next.
>
> **“Why should anyone care if a protein design algorithm is worse at designing a robot controller?”, These problems share little structure, “no free lunch” applies… Existence of a master algorithm?**
>
> We agree with the reviewer that the concern for “no free lunch” is valid such that there does not exist an algorithm that works well for all possible offline MBO tasks. However, looking at the history of the development of machine learning and deep learning methods, we find that researchers have indeed come up with algorithms, like SVM, random forests, gradient descent and neural networks, that are generally applicable in many real-world domains that vary greatly in terms of structure. Even if these methods by themselves are not the most powerful way of solving a particular problem, they often serve as foundations for building problem-specific solutions. Therefore, we believe that our benchmark can also help researchers develop principles for devising offline MBO methods widely applicable to many real-world problems. While these principles may not be instantiated in the same way across each domain, and could use different implementation choices, we believe our benchmark will still be useful for obtaining such general-purpose principles and hence for developing performant offline MBO methods.
>
>
> **“GFP task: Why is the GP model used as groundtruth? Other choices?”**
>
> For the GFP task, we choose the GP oracle following the convention of prior work (Brookes et al. 2019). Following the reviewer’s suggestion, we have now added a new task, labeled GFP-v1, that adapts the 12-layer Transformer model provided by the TAPE framework (Rao et al. 2019) trained to map from amino acid sequences to protein fluorescence measured in brightness. The model is pre-trained using BERT on a dataset adapted from the Pfam protein database, and fine-tuned for fluorescence prediction on the original GFP dataset.
>
> We evaluated all algorithms on GFP-v1, and the relative performance between algorithms remains almost the same, as shown in Tables 2 and 3, with a rank correlation of 0.937 compared to the rankings of methods under GFP-v0. This result indicates that the GP model in GFP-v0, although less accurate than the state-of-the-art model, is sufficient for evaluating offline MBO algorithms. We will also look into creating a version of the task using the Rosetta software suite in the final version of the paper.

---

> > ### Author Response · Authors · 2020-11-19
> > **Author Response (Part 2 of 2): 5 new baselines, code structure and API**
> >
> > **Additional Baselines**
> >
> > To address the reviewer’s concerns, as suggested by the reviewer, we have incorporated **5** new baselines in Tables 2 and 3 that include a genetic algorithm, CMA-ES; variants of gradient ascent on uncertainty-aware objectives obtained from an ensemble of learned models of the objective, Bayesian optimization based on Gaussian processes, as well as a variant of the method from Angermueller et al. (ICLR 2020). If the reviewer suggests or can refer us to other state-of-the-art offline MBO methods, we will be happy to include them in our benchmark.
> >
> >
> > **Code structure, Design-bench API and Algorithm code**
> >
> > We apologize for the delay. The design-bench and our reference implementation of offline MBO algorithms are available on the website (https://sites.google.com/view/design-bench). As for the API, we want to clarify that we design the APIs to be minimal intentionally. We recognize that different MBO algorithms are structured differently and require different diagnostic metrics. Researchers would also want to use different software frameworks and packages to develop their algorithms. Therefore, it is usually impossible to encapsulate all these needs under a fixed set of APIs. We also note that successful machine learning benchmarks in the past such as ImageNet and OpenAI Gym often have minimal APIs, and do not handle the evaluation or sanity check for the users.
> >
> > **Gradient methods perform poorly in MBO**
> >
> > We have updated the list of references for gradient methods performing poorly in MBO. The results in Brookes et al. 2020 show that the Gomes-Bomberelli method, a gradient-based method, does not perform well for GFP.  Kumar and Levine, 2019 also show that gradient-based methods can be poor in solving MBO tasks over images.
> >
> >
> > We request the reviewer to kindly revisit the paper in the light of these revisions and please let us know if additional modifications to the paper are needed.

---

> > > ### Comment · AnonReviewer2 · 2020-11-25
> > > **Paper has been improved, but several shortcomings remain.**
> > >
> > > I thank the authors for their efforts to address my concerns. I believe the paper has been improved.
> > >
> > > Several problems are remaining, I order them for "addressability", most to least:
> > >
> > > (1) The links provided for the code are still broken, i.e. the actual code under design bench directory is missing. I attempted download multiple times and with different browsers. I cannot approve the paper for acceptance without reading the code.
> > >
> > > (2) I find that the API is "minimal" to a fault, currently all it does is to load the x,y labels for you. That's not at all sufficient or equivalent in abstraction to OpenAI gym. This is the main contribution of the paper. You cannot be in a regime where different people attempt to run CbAS in the same environment, and get different results from those in the paper. There should be a way to directly get the "benchmark results" and plug in another algorithm and compare.
> > >
> > > (3) REINFORCE is not DynaPPO (as CEM is not CbAS). DynaPPO claims to beat CbAS in protein design. I think the authors should strive to reproduce it exactly instead of other baselines.
> > >
> > > ..........
> > >
> > > Side point: It is no longer the case that Gradient Ascent is generally a better algorithm. It is hard for me to contextualize these results without the others including any of the top performing algorithms in each domain.

---

> > > > ### Author Response · Authors · 2020-11-25
> > > > **Thank you for the response! Addressing these concerns below.**
> > > >
> > > > We thank the reviewer for their constructive feedback, going over the responses, and participating in a discussion with us.
> > > >
> > > > **Link to code** At the outset, we sincerely apologize for the broken links to the folder inside the code zip file, which we didn’t realize earlier. This has now been fixed, and the code for both the tasks and the baselines is available at:
> > > >
> > > > Benchmark Code (Design-bench): https://drive.google.com/file/d/1A72kut9RDBZHVCWVzu0c7NSTNOZGuHSB/view?usp=sharing
> > > >
> > > > Baselines Code (Design-baselines): https://drive.google.com/file/d/1PkuJHe5NUQAQRXe2sIzGiIJp-XdP3PNP/view?usp=sharing
> > > >
> > > > > API is "minimal" to a fault, currently, all it does is to load the x,y labels for you. That's not at all sufficient or equivalent in abstraction to OpenAI gym. There should be a way to directly get the "benchmark results" and plug in another algorithm and compare.
> > > >
> > > > **Direct way to get benchmark results and comparison.**  We would first like to point out that we have released an open-source reference implementation of algorithms along with the benchmark called design-baselines (the code is also available on the website). Any algorithm developer can directly run the code to reproduce existing results in our paper or use the code as a foundation to develop future algorithms. For example, CbAS is located in file design-baselines/design-baselines/design_baselines/cbas/\_\_init\_\_.py, and design-baselines/design-baselines/design_baselines/cbas/experiments.py. The function which runs cbas on gfp_v1 is design-baselines/design-baselines/design_baselines/cbas/experiments.py:gfp_v1. Therefore, we believe the design-baselines codebase addresses the reviewer's concern about reproducing results and scaffolding the evaluation of future algorithms.
> > > >
> > > > **Minimality of API and analogy to OpenAI gym.** We will be happy to incorporate any suggestions that the reviewer has into the design of the API. However, to the best of our knowledge, we believe that if we were to draw a comparison between Design-Bench and OpenAI gym: OpenAI gym also just provides us with the ability of obtaining a y (reward value) for a given x (input action), and has additional machinery to handle the sequential nature of the RL problem. We believe that our API does something similar, but is significantly simpler since we consider only 1-step optimization problems.  We would also like to point out that our design of API decouples the benchmark from the implementation and evaluation of the algorithms, much like how OpenAI gym is decoupled from OpenAI baselines or other RL libraries. This allows us to give researchers full freedom to develop future algorithms their own way while also providing an optional reference implementation if they want to use it.
> > > >
> > > > > REINFORCE is not DynaPPO (as CEM is not CbAS)...I think the authors should strive to reproduce it exactly instead of other baselines.
> > > >
> > > > We agree with the reviewer that REINFORCE is not DynaPPO and apologize if our response implied this. We compared to REINFORCE since some of our tasks (e.g., neural network weight optimization in HopperController-v0) cannot be factorized sequentially, and in such one step settings we used REINFORCE with models of the objective function chosen in a similar way as DynaPPO’s cross-validation threshold. To address the reviewer’s concerns, we will add the exact DynaPPO method in the final version of this paper and the benchmark.
> > > >
> > > > > Side point: gradient ascent
> > > >
> > > > The reviewer is right that there are some other methods, including variants of gradient ascent, that outperform the gradient descent baseline in Tables 2 and 3. We are not claiming in the paper that it is the best algorithm to solve offline MBO problems. Instead, we are presenting it as a surprising finding that gradient ascent -- a very simple baseline -- outperforms generative modeling methods on a number of tasks.
> > > >
> > > > We thank the reviewer for going over our paper and response and providing constructive feedback. We hope that our responses above address the concerns raised by the reviewer. We would appreciate it if the reviewer can let us know if any concerns are still remaining.

---

> ### Author Response · Authors · 2020-11-22
> **Discussion**
>
> Dear reviewer,
>
> Please let us know if our response below addresses the concerns raised in your review. We will be happy to clarify these or other concerns more.

---

> ### Author Response · Authors · 2020-11-24
> **Request for discussion**
>
> Dear Reviewer,
>
> Thank you for the constructive feedback on our paper. As we near the end of the discussion period (in less than 24 hours now), we are hoping to hear if our responses below address the concerns in the review. If there is anything that is not addressed, we are happy to address it in the remaining time. We would appreciate it if you can tell us if there are any other concerns.
>
> Thanks,
> Authors

---

### Official Review · AnonReviewer4 · 2020-10-29
**A meaningful benchmark for offline optimization**

**Rating:** 7
**Confidence:** 4

**Review:**

I liked the paper overall. The motivation of the paper is clear, and given that offline ML based optimization is beginning to take traction, this is a good time to set up benchmarks and evaluation metrics. The variety of domains considered, with careful consideration of complexity, are good characteristics of the benchmark.

The simple baseline considered makes sense and challenges the community to develop algorithms that can generalize to different problem characteristics.

A few suggestions for improving the paper:
- The hyper-parameters for each algorithm need not be the same for each task. It is possible that tuning the hyper-parameters for particular tasks improves the performance of the algorithms.
- As the paper points out, one cannot evaluate these algorithm at scale as they require real world experimentation. An important aspect that is not discussed in the paper is offline evaluation of the optimization algorithms. In a practical application, we need to know the efficacy of an algorithm for a particular task before they are deployed on the real task.
- None of the tasks considered have constraints on the problem. This is especially challenging for model based methods in continuous design space.
- The algorithms implemented only consider model based methods. The more popular traditional methods such as genetic algorithms and mixed integer programming would be good to compare against. ML methods will only be adopted if they can beat existing established methods.

---

> ### Author Response · Authors · 2020-11-19
> **Author Response: Hyperparameter Tuning, Offline Evaluation, Constraints and Other Baselines**
>
> We thank the reviewer for their constructive feedback and are glad that they found our benchmarking effort interesting and timely. We have made revisions to the paper to elaborate on our hyperparameter selection procedure (Appendix F) and added traditional methods (genetic algorithms, REINFORCE, Bayesian optimization) in Tables 2 and 3. These changes in the paper are marked in blue. We now answer specific questions.
>
> **“The hyperparameters for each algorithm need not be the same for each task”**
> We agree with the reviewer that hyperparameters for each algorithm need not be the same for each task. However, we would like to point out that our results in Tables 2 and 3 do not use identical values of different hyperparameters across different tasks, but use identical offline procedures to select hyperparameters. For instance, we tune the $\beta$ value for the $\beta-$VAE in CbAS until samples from the VAE prior resemble a held-out validation set. We have also added a discussion of the workflow for hyperparameter tuning for each method in Appendix F.
>
> Certainly tuning hyperparameters more can improve performance of each method, however, in the offline MBO setting, there is no access to the groundtruth function that is being optimized. This makes it challenging to tune hyperparameters since we do not have access to a groundtruth evaluation metric during training.
>
> **“Offline Evaluation of Optimization Algorithms not discussed”**
> We agree with the reviewer that this indeed an important problem, especially in cases where we do not have access to a simulator for evaluating a design method. We do not consider offline evaluation in this paper and we have added a discussion of offline evaluation of MBO algorithms as a subject of future work.
> In this paper, we are only interested in evaluating offline MBO algorithms based on their groundtruth, online performance.
>
> **“None of the tasks have constraints on the problem”**
> We note that under our formulation, constraints can be encoded by modifying the objective function to take extremely low values for invalid inputs that do not satisfy the constraints. For a number of tasks in our benchmark, such as the morphology design tasks, the space of valid/stable morphologies is constrained to lie within a hypercube (see the design-bench code) -- everything outside this region is invalid, and optimization needs to find a solution within this set. We are happy to add more tasks with constraints if the reviewer has any suggestions.
>
> **“Genetic algorithms and mixed-integer programming baseline methods”**
> We have updated the paper to include 5 new baselines, including CMA-ES, which is a genetic algorithm; Bayesian Optimization; REINFORCE, which is an adaptation of a method from AngerMueller et al. ICLR 2020, and two variants of gradient ascent using ensembles of learned models of the objective function in Table 2. To the best of our knowledge, mixed integer programming methods require access to the functional form of the ground truth objective function, which cannot be directly applied in our setting which only assumes sample access to the grountruth function.
>
>
> We request the reviewer to kindly revisit the paper in the light of these revisions and please let us know if additional modifications to the paper are needed.

---

### Official Review · AnonReviewer1 · 2020-10-29
**Official Blind Review #1**

**Rating:** 5
**Confidence:** 2

**Review:**

 ##########################################################################

Summary:

This paper proposes a benchmark suite of offline model-based optimization problems. This benchmark includes diverse and realistic tasks derived from real-world problems in biology, material science, and robotics contains a wide variety of domains, and it covers both continuous and discrete, low and high dimensional design spaces. The authors provide a comprehensive evaluation of existing methods under identical assumptions and get several interesting takeaways from
the results. They found there exists surprising efficacy of simple baselines such as naive gradient ascent, which suggests the need for careful tuning and standardization of methods in this area.

##########################################################################

Pros:

1. This paper tackles a valuable problem of benchmarking model-based optimization approaches. It will provide some insights in future algorithmic development

2. The paper is well written. They present the significance of model-based optimization, and clearly describe the problems, challenges and considerations of model-based optimization tasks. They conduct further analysis and discussions on the experimental results and find several interesting takeaways.

##########################################################################

Cons:

1. Contribution is limited. There are no new ideas proposed and no significant findings revealed. Maybe the authors can look deeper into the simple takeaways and move forward to get more insights and draw some generalized conclusions.

2. In continuous control tasks such as HopperController, I think we cannot get a truly policy without trajectory data. I do not believe the datapairs of the weights of a neural network controller and the corresponding return values can be directly used to learn useful knowledge that contributes to calculate a reasonable policy. It is hard to get a generalizable function that maps from weights of a controller network to resulting values.

---

> ### Author Response · Authors · 2020-11-19
> **Author Response: Addressed Novelty and HopperController Task**
>
> We thank the reviewer for the detailed comments and constructive suggestions. The main reservations in the review center on the fact that no new ideas have been proposed, but we would like to emphasize that the benchmark of offline MBO problems we propose is in itself a new idea. Our experiments also provide new insights that can be used to guide future research on offline MBO, such as the fact that an extremely simple gradient ascent baseline can be performant in offline MBO problems when simple tricks such as normalization of the inputs and objectives is performed. We address the reviewer’s concerns in detail below.
>
> **“No new idea, no significant findings revealed”**
> First of all we want to clarify that the benchmark we propose is by itself a “new idea”, since such a benchmark doesn’t exist in the field of offline MBO. The standardization of tasks and evaluation protocols could greatly assist the development of future offline MBO algorithms, as benchmarks have done in other fields such as computer vision, natural language processing and reinforcement learning (Brockman et al. 2016).  Beyond the benchmark, we also discover the surprising effectiveness of a simple gradient ascent baseline, when instantiated with normalized designs and objective spaces, which is surprising in the light of prior works like Brookes et al. 2019 and Kumar and Levine 2019, as also pointed out by Reviewer 4. This also dictates that understanding the right design decisions with MBO methods, and in particular, with gradient ascent methods has the potential to give rise to simple and effective offline MBO algorithms.
>
> **“HopperController, cannot get a reasonable policy without trajectory data”**
> We agree with the reviewer that directly operating in the policy parameter space without access to the transition data from the MDP is not the best way of solving the policy optimization problem. However we do want to emphasize that the purpose of our benchmark is to evaluate offline MBO algorithms, not offline RL algorithms. The problem of optimizing policy parameters for return is a legitimate high-dimensional offline MBO problem, and although many algorithms fail on this task, the results from our gradient ascent baseline demonstrate performance on this problem can be improved over the best possible datapoint in the dataset -- note in Table 1 that the gradient ascent baseline obtains a performance of about 2x the best possible controller in the dataset, which indicates that this task is not impossible and provides room for improvement.

---

> ### Author Response · Authors · 2020-11-22
> **Discussion**
>
> Dear reviewer,
>
> Please let us know if our response below addresses the concerns raised in your review. We will be happy to clarify these or other concerns more.

---

> ### Author Response · Authors · 2020-11-24
> **Request for discussion**
>
> Dear Reviewer,
>
> Thank you for the constructive feedback on our paper. As we near the end of the discussion period (in less than 24 hours now), we are hoping to hear if our responses below address the concerns in the review. If there is anything that is not addressed, we are happy to address it in the remaining time. We would appreciate it if you can tell us if there are any other concerns.
>
> Thanks,
> Authors

---

### Decision · Program_Chairs · 2021-01-07
**Final Decision**

**Decision:**

Reject

**Comment:**

This paper proposes a benchmark suite of offline model-based optimization problems. This benchmark includes diverse and realistic tasks derived from real-world problems in biology, material science, and robotics contains a wide variety of domains, and it covers both continuous and discrete, low and high dimensional design spaces. The authors provide a comprehensive evaluation of existing methods under identical assumptions and get several interesting takeaways from the results. They found there exists surprising efficacy of simple baselines such as naive gradient ascent, which suggests the need for careful tuning and standardization of methods in this area, and provides a test bed for algorithms that try to solve this challenge. However, most reviewers agreed that a more in-depth analysis and insightful explorations for the RL experiment results will help readers understand why their method has superiority even without trajectory data, and  that the paper needs another revision before being accepted. Therefore, I recommend rejection although all reviewers agreed that the tasks is very interesting and a good start.